# VISUALLY-AUGMENTED PRETRAINED LANGUAGE MODELS FOR NLP TASKS WITHOUT IMAGES

## ABSTRACT

Although pre-trained language models (PLMs) have shown impressive performance by text-only self-supervised training, they are found lack of visual semantics or commonsense, *e.g.,* sizes, shapes, and colors of commonplace objects. Existing solutions often rely on explicit images for visual knowledge augmentation (requiring time-consuming retrieval or generation), and they also conduct the augmentation for the whole input text, without considering whether it is actually needed in specific inputs or tasks. To address these issues, we propose a novel **v**isually-**a**ugmented fine-tuning approach that can be generally applied to various PLMs or NLP tasks, **w**ithout using any retrieved or generated **i**mages, namely **VAWI**. Specifically, we first identify the visually-hungry words (VH-words) from input text via a token selector, where three different methods have been proposed, including syntax-, attention- and learning-based strategies. Then, we adopt a fixed CLIP text encoder to generate the visually-augmented representations of these VH-words. As it has been pre-trained by vision-language alignment task on the large-scale corpus, it is capable of injecting visual semantics into the aligned text representations. Finally, the visually-augmented features will be fused and transformed into the pre-designed visual prompts based on VH-words, which can be inserted into PLMs to enrich the visual semantics in word representations. We conduct extensive experiments on ten NLP tasks, *i.e.,* GLUE benchmark, CommonsenseQA, CommonGen, and SNLI-VE. Experimental results show that our approach can consistently improve the performance of BERT, RoBERTa, BART, and T5 at different scales, and outperform several competitive baselines significantly. Besides, the generated visual prompts of our framework can also be used for parameter-efficient tuning, which boosts the performance of T5-3B. We will make our code, data, and models publicly available.

## 1 INTRODUCTION

Recent years have witnessed the success of pre-trained language models (PLMs), such as GPT-3 (Brown et al., 2020) and T5 (Raffel et al., 2020), in a variety of natural language process (NLP) tasks. Since these PLMs are mostly trained on text-only corpus via self-supervised pre-training, they have been shown lack of visual commonsense (Liu et al., 2022) and real-world knowledge (Zhang et al., 2022). As a result, PLMs can't well solve visually related language tasks [1], *e.g.,* answering the color and size of common things, especially those requiring complex commonsense knowledge.

To alleviate this problem, existing works mainly enhance PLMs by infusing the visual information. Typically, given a text input, these studies firstly augment the visual information from retrieved or generated images about the input and then leverage their visual representations to improve PLMs on NLP tasks. Such an approach leads to *visually-augmented pre-trained language models (VaLMs)*, where they adopt either visually-augmented pre-training (Tan & Bansal, 2020; Wang et al., 2022) or visually-augmented fine-tuning techniques (Lu et al., 2022). Despite the effectiveness, there are two major shortcomings in these methods. First, it is very costly to retrieve or generate high-quality images that are paired with the input. These methods often rely on pre-learned complementary retrievers or generators, and also require time-consuming inference for obtaining the images, which

---

[1] In this work, we mainly consider the text-only NLP tasks that may need the visual information as complementary, but not the visual-language tasks with images.

largely limits the applicability of this approach. Second, as the augmented visual information comes from retrieved or generated images, it is inevitable to involve irrelevant or redundant visual information. If simply integrating the augmented visual information, the original text representations might be affected or even "polluted". Increasing evidence shows that the visual information is not always useful for NLP tasks (Dai et al., 2022), and sometimes leads to performance degradation.

Considering these issues, we aim to develop a more efficient and effective way to visually augment the PLMs and the solution is twofold:

• Firstly, we don't explicitly produce (retrieve or generate) the images but instead generate visually-aligned representations of the text on-the-fly. Recent studies (Radford et al., 2021; Jia et al., 2021) have shown that the vision-language pre-trained models (VL-PTMs) can well learn the alignment between the representations of texts and images from large-scale text-image pairs. Thus, our idea is to employ the output representations of a text from VL-PTMs' text encoders as a surrogate for the visual representations of related images.

Such a way is simple and efficient: we can only keep the text encoder of a VL-PTM to produce the visually-aligned representations of texts, getting rid of the complicated image retrieval or generation process. It is widely recognized that there is a large semantic gap between different modalities (Liang et al., 2022). Our method can alleviate this issue to some extent since the visual augmentations are derived from the text representation itself, *i.e.,* visually-aligned text representations from VL-PTMs.

• Secondly, instead of directly feeding visual augmentations into the PLM, we propose to use the augmented visual information only when it actually needs. In fact, for a text input of a NLP task, PLMs are not always hungry for the visual background knowledge to effectively understand it, especially for visually-irrelevant expressions. Unlike previous works which inject visual information into a text (Tan & Bansal, 2020; Wang et al., 2022) from the whole, we consider identifying *visually-hungry words* (those that require visual knowledge to derive complete semantics) from the text input, and only infuse the visual augmentations through these trigger words. We conduct visual augmentations at the word level, because it is more flexible and controllable, considering the augmented information is often irrelevant or noisy.

To this end, in this paper, we propose a general Visually-Augmented fine-tuning approach to improving PLMs for NLP tasks Without Images, namely **VAWI**. Our approach consists of three ingredients, namely visually-hungry words extraction, visual knowledge augmentation, and visually-enhanced fine-tuning. Given the text input from a NLP task, we first extract the visually-hungry words (VH-words) from the input sentence. As the annotations of VH-words are generally unavailable, we propose three strategies to automatically extract the VH-words, relying on the syntax trees, attention distributions of the VL-PTMs' text encoder, and an adaptive learnable module, respectively. Then, based on the extracted VH-words, we leverage the text encoder of CLIP (Radford et al., 2021) (being fixed in our approach), a VL-PTM that has been pre-trained on millions of text-image pairs, to encode the VH-words for obtaining their visually-aligned representations. Finally, we design visually-enhanced fine-tuning strategies to infuse the visually-aligned representations into PLMs. For small PLMs, we directly incorporate these visual representations to enrich the word embeddings and fine-tune the parameters of the PLM and our approach. For large-scale PLMs, we also propose a parameter-efficient prompt-tuning strategy that only tunes very few parameters in our approach, with the PLM being frozen.

To summarize, our approach provides an adaptive, flexible and efficient way to leverage visual information for enhancing text-based PLMs. For verifying the effectiveness of our framework **VAWI**, we test it on four PLMs (*i.e.,* BERT, BART, RoBERTa, and T5) at different scales (*i.e.,* 110M, 340M, 3B), and conduct extensive experiments in natural language understanding, commonsense reasoning, and text generation tasks. Experimental results show that our **VAWI** can boost the performance of these PLMs significantly, *i.e.,* 3.11%, 2.54%, and 2.16% absolute improvements on the commonsenseQA task using RoBERTa-base, RoBERTa-large, and T5-3b, respectively. Besides, **VAWI** can outperform (or be on par with) several competitive baselines that adopt complicated visually-augmented methods. Additionally, **VAWI** further improves the performance of a VL-PTM (*i.e.,* ALBEF(Li et al., 2021)) on the cross-modal reasoning task by enhancing its text-encoder.

## 2  RELATED WORK

In this section, we illustrate related work from the following three perspectives, namely pre-trained language models, vision-language pre-trained models and visually-augmented language model.

**Pre-trained Language Models.**  Recent years have witnessed the success of pre-trained language models (PLMs) (Devlin et al., 2019; Radford et al., 2019; Lewis et al., 2020; Raffel et al., 2020). After pre-trained on large-scale corpus, PLMs can be fine-tuned on multiple NLP tasks and achieve remarkable performance. Typically, PLMs incorporate the Transformer architecture (Vaswani et al., 2017) and require large-scale corpus. With the increasing scale of the model parameters and pre-training corpus, PLMs can even achieve human performance (Devlin et al., 2019; Radford et al., 2019). However, since PLMs are just pre-trained with text-only data, they may suffer from the reporting bias problem (Gordon & Van Durme, 2013; Paik et al., 2021; Zhang et al., 2022), where the frequency distribution of visual commonsense in the text may not reflect the real-world distribution of the commonsense. Existing works have also found that such a problem can not be addressed by enlarging the model or pre-training corpus (Paik et al., 2021; Zhang et al., 2022). In this work, we aim to reduce the influence of the reporting bias problem on PLMs by adding visual knowledge during fine-tuning.

**Vision-Language Pre-Trained Models.**  To better accomplish the vision-language tasks, vision-language pre-trained models (VL-PTMs) (Su et al., 2019; Lu et al., 2019; Li et al., 2021) become a hot point in recent years, which require large-scale image-text pairs for pre-training. Existing VL-PTMs fall into two categories based on the way of modeling vision-language interaction. The first category of models (Su et al., 2019; Lu et al., 2019; Li et al., 2021) adopts an explicit vision-language interaction layer to fuse the text embeddings and image features using a single-stream or dual-stream network structure. These models are more suitable for the tasks that need to capture fine-grained semantic interactions between vision and language (*e.g.,* NLVR, VQA, and VE), and have achieved remarkable performance on them (Li et al., 2021; Alayrac et al., 2022). The second category of models (Radford et al., 2021; Jia et al., 2021) incorporates separate encoders to independently model the vision and language information, and relies on pre-training tasks (*e.g.,* cross-modal contrastive learning) to align their representations into the same space. Such a way is capable of producing enriched single-modal representations and has achieved remarkable performance on both vision and image-text retrieval tasks (Radford et al., 2021).

**Visually-Augmented Language Model.**  Visually-augmented pre-trained language model (VaLM) (Wang et al., 2022) has become an emerging research topic that aims to introduce the visual information into PLMs for alleviating the reporting bias problem. Existing VaLMs can be categorized into visually-augmented pre-training and fine-tuning methods. Visually-augmented pre-training approaches (Tan & Bansal, 2020; Wang et al., 2022) have to continually pre-train the PLMs with the visual information. They mostly retrieve the visual information related to input tokens or sentences from a large-scale images dataset and revise the masked language model task for better capturing the visual semantics. Visually-augmented fine-tuning methods (Lu et al., 2022) introduce the visual information into PLMs during the fine-tuning stage. These methods also leverage the image retrieval or generation models to augment the visual information and design a special fusion module to inject it into PLMs. However, existing VaLM approaches mostly need to retrieve or generate visual information for utilization. Such a way is not only time-consuming, but may also involve unrelated or noisy information into PLMs, leading to performance degradation. In this work, we aim to first detect the visually-hungry words from the text, and then utilize a VL-PTM to generate their visually-aligned representations without the usage of external images or generation models. As a comparison, our approach is more flexible and efficient to leverage the visual information for enhancing text-based PLMs.

## 3  METHOD

In this section, we firstly introduce the task setting, and then describe our proposed visual augmentation approach for infusing visual knowledge into PLMs during fine-tuning.

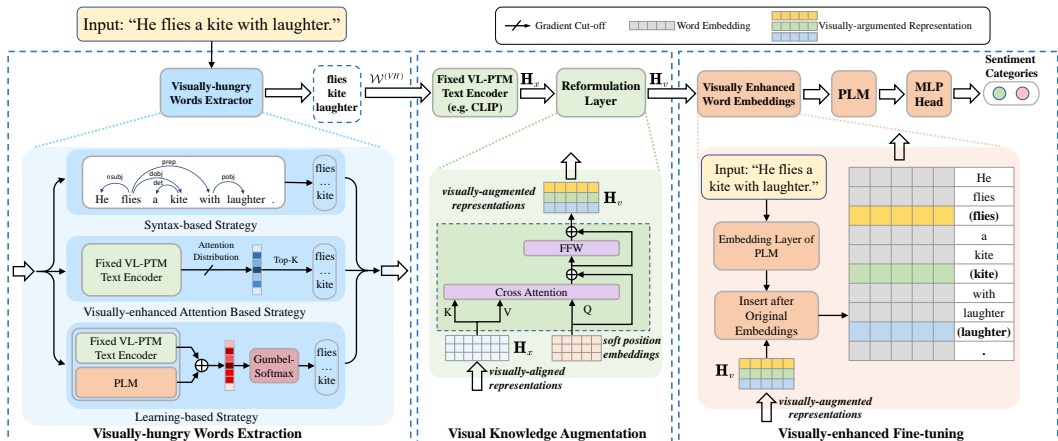

Figure 1: The illustration of our VAWI approach, consisting of visually-hungry words extraction, visual knowledge augmentation and visually-enhanced fine-tuning.

## 3.1 TASK SETTING AND SOLUTION OVERVIEW

This work aims to improve the fine-tuning performance of pre-trained language models (PLMs) on NLP tasks by leveraging the related visual information without images. For a NLP task, a set of $n$ labeled texts $\{\langle x_i, y_i \rangle\}$ are available, where $x_i$ is the $i$-th text data consisting of a sequence of words, denoted as $x_i = \{w_1, w_2, ..., w_m\}$, and $y_i$ is the ground-truth output. Note that we don't specify the form of $y_i$, which can be a discrete label (classification), a continuous value (regression) or a text sequence (generation), since our approach is generally applicable to various settings (either natural language understanding or generation tasks).

To solve the target task, we assume that a text-based PLM is given (either for understanding or generation). Let $f$ denote a PLM parameterized by $\theta_{\text{PLM}}$ that has already been pre-trained on general-purpose large-scale text data. We mainly focus on the encoder of the PLM, which learns the input embedding via the "[CLS]" token. Given the labeled training data, we can train the PLM using specific loss function (*e.g.,* cross-entropy loss) and further solve the target task. However, existing works (Tan & Bansal, 2020; Zhang et al., 2022) have revealed that PLMs may be unaware of visual knowledge that is not explicitly mentioned in the pre-trained text-only data (*e.g.,* the shape of coins and the color of the sky), leading to the lack of world commonsense and generating wrong statements.

In this work, we focus on devising an efficient and effective way to infuse such visual knowledge into PLMs during fine-tuning. Our approach is based on *visually-hungry words* (abbreviated as *VH-words*), which require the visual information to derive complete semantic representations. The overall illustration of our approach is shown in Figure 1, which consists of three important ingredients, namely VH-words extraction, visual knowledge augmentation, and visually-enhanced fine-tuning. Given the input text $x_i$ and its label $y_i$, we first detect and extract a set of VH-words. Then, we adopt a visual knowledge augmentation module to enhance the visual background knowledge of their tokens and generate their visually-aligned representations. Finally, we infuse the visually-aligned text representations into the PLM to improve its fine-tuning performance, where we consider both the general fine-tuning of small PLMs and the parameter-efficient fine-tuning of large-scale PLMs.

## 3.2 VISUALLY-HUNGRY WORDS EXTRACTION

In our approach, visually-hungry words (VH-words) are the trigger units for visual augmentations, requiring visual knowledge for deriving complete semantic representations. For example, some visually-related words (*e.g.,* color, shape and object) may not be well learned by the text-only PLM. Therefore, we propose to first detect the VH-words from the input text, and then inject the proper visual knowledge that they are hungry for into the PLM. However, the annotations about VH-words are generally not available in NLP datasets. To address this problem, we devise three different

strategies to extract the VH-words from the input text, including two feature-based strategies based on syntax tree and attention distribution of PLMs, and a learnable model-based strategy.

**Syntax-based Strategy.** In natural language, entity words and descriptive words usually convey more visual semantics than others. For example, for the sentence "*He is eating a green apple*", where underlined words are more related to visual semantics. Such words are mostly nouns or adjectives in the input text, which can be detected by syntactic analysis. Therefore, we design a rule-based strategy that leverages the syntactic information of the words in $s$ for VH-words extraction. Concretely, we first delete all stop words in a text and then adopt an open-resource toolkit SPACY [2] to convert the input text into a syntax dependency tree. Based on the syntax tree, we extract the words that have a particular part of speech (POS), *e.g.,* nouns or adjectives, as the VH-words denoted by $\mathcal{W}^{(VH)}$. In this way, we can efficiently extract the VH-words from input text by using a fast parser toolkit. However, as such a strategy mainly relies on the syntax information, it may select some nouns or adjectives that are not explicitly related to visual semantics (*e.g.,* sweet and bitter). In the following, we will introduce two improved strategies to alleviate this problem.

**Visually-enhanced Attention Based Strategy.** The attention-based strategy utilizes the attention distribution of a VL-PTM to detect the VH-words. Since VL-PTMs (Radford et al., 2021) are pre-trained on large-scale image-text pairs, their text encoders are able to focus more on the words corresponding to some specific visual concepts in an image, which are likely to be VH-words. Inspired by this idea, we can use the attention scores calculated by the text encoder of VL-PLMs to select the VH-words. Specifically, we adopt the text encoder of CLIP (Radford et al., 2021), a VL-PTM that has been pre-trained on millions of image-text pairs, to help extract the VH-words. As CLIP adopts an autoregressive GPT-2 model as the text encoder, we calculate the average attention scores between each token and the "[EOS]" token on the last multi-head self-attention layer, denoted as $s_{w_i}$. Then, we select the top-$K$ ranked words according to $\{s_{w_i}\}$ as the VH-words $\mathcal{W}^{(VH)}$. In this way, we can leverage the intrinsic knowledge encoded by the VL-PTM to extract VH-words.

**Learning-based Strategy.** Considering that diverse PLMs and NLP tasks may be hungry for the different complementary visual information, we devise a learning-based strategy that can adaptively extract VH-words according to task requirements. Concretely, we add a parameterized VH-words extractor layer for the PLM, which can be updated by gradient-based optimization algorithms to fit the need for some specific task. Given the input text $x_i$, we first leverage the PLM and a text encoder of a VL-PTM (*i.e.,* CLIP (Radford et al., 2021)) to produce the contextualized representations of the contained words in $x_i$. Then, we concatenate the representations of each word from the two models and utilize a MLP layer to project it into the score $s_{w_i}$:

$$s_{w_i} = \text{MLP}([\mathbf{h}_{w_i}^{(P)}; \mathbf{h}_{w_i}^{(V)}]) \tag{1}$$

where $\mathbf{h}_{w_i}^{(P)}$ and $\mathbf{h}_{w_i}^{(V)}$ are the output word representations from the PLM and VL-PTM, respectively, and scores $s_{w_i}$ are calculated by the learned model based on the supervision information from downstream tasks. Based on the scores of all words, we incorporate the gumbel-softmax function (Jang et al., 2016) to extract the top-$k$ words as the VH-words in a differentiable way. In this way, the gradients of the fine-tuned tasks can be back-propagated to the extractor layer, which learns to adaptively select the more suitable VH-words.

## 3.3 VISUAL KNOWLEDGE AUGMENTATION

Existing works (Lu et al., 2022; Wang et al., 2022) mainly utilize image retrieval or generation module to augment related visual knowledge. Such a way is time-consuming and may also involve noisy images that will influence the task performance. Inspired by recent works that show the effective visual-language alignment in VL-PTMs (Radford et al., 2021; Li et al., 2021), we utilize the visually-aligned text encoders as the visual augmentation representations of VH-words. As the text encoders have been aligned to the image encoders during pre-training on large-scale image-text pairs, their output textual representations can be used as surrogates of visual augmentations based on real images related to the input text. As will be shown in experiments (Section 4), this approach is not only efficient but very effective for downstream NLP tasks.

---

[2] https://spacy.io/

Based on the extracted VH-words, we first add a prefix text in the image caption style before the VH-words, *e.g.,* "*a photo of:* ", to compose the input text $x'$. Then, we utilize the text encoder of CLIP (Radford et al., 2021) to encode $x'$ and obtain the contextualized word representations as the visually-aligned representations $\mathbf{H}_x \in \mathbb{R}^{k \times d}$, where $k$ is the sequence length of $x'$ and $d$ is the embedding size. To effectively reserve the visual semantics of VH-words, we obtain the visual representation of each VH-word in $\mathcal{W}^{(VH)}$. For example, given a sentence "*a green apple and a big orange*", we enhance the visual knowledge of underlined words with visually-aligned representations $\mathbf{H}_x$, respectively. Therefore, we incorporate a reformulation layer to aggregate the visually-aligned representation $\mathbf{H}_x$ into the visually-augmented representations of these VH-words. However, the positions of the VH-words vary from sentence to sentence. Therefore, we design a position-aware attention mechanism in the reformulation layer to inject position information into $\mathbf{H}_x$ for obtaining the visual representation of each VH-word via $\mathbf{H}_x$. Specifically, we leverage a soft position embedding matrix $\mathbf{E} \in \mathbb{R}^{l \times d}$ to reserve the position information of VH-words, where $l$ is the number of VH-words, and perform cross-attention between it and the visual representations as:

$$\mathbf{Q} = \mathbf{E}, \quad \mathbf{K} = \mathbf{H}_x \mathbf{W}^K + \boldsymbol{b}^K, \quad \mathbf{V} = \mathbf{H}_x \mathbf{W}^V + \boldsymbol{b}^V, \tag{2}$$

$$\mathbf{H}_v = \text{softmax}(\frac{\mathbf{Q}\mathbf{K}^\top}{\sqrt{d}})\mathbf{V}, \tag{3}$$

$$\mathbf{H}_v^\top = \{\mathbf{h}_1, \mathbf{h}_2, ..., \mathbf{h}_l\}, \tag{4}$$

where $\mathbf{h}_i \in \mathbb{R}^d$, $\mathbf{K}, \mathbf{V} \in \mathbb{R}^{k \times d}$. $\mathbf{H}_v \in \mathbb{R}^{l \times d}$ is the obtained visually-augmented representations of VH-words, which can be leveraged for augmenting the visual knowledge of the PLM during fine-tuning. $\mathbf{h}_i$ is the visual representation of the $i$-th VH-word in $\mathcal{W}^{(VH)}$, which are sorted by the position of VH-word in $x'$. Note that in Eq. 2, we only use the position information to set the *query* matrix $\mathbf{Q}$, which is more efficient than standard self-attention (Vaswani et al., 2017), and the visual semantics are mainly captured and injected through the *key* and *value* matrices.

### 3.4 VISUALLY-ENHANCED FINE-TUNING

After obtaining the visually-augmented representations of VH-words (*i.e.,* $\mathbf{H}_v$ in Eq. 4), we propose a visually-enhanced fine-tuning strategy to inject the captured visual knowledge. Here, we consider two cases: (1) full-parameter fine-tuning for small PLMs, and (2) parameter-efficient prompt-tuning for large-scale PLMs. Before introducing the learning method, we simply review the parameters to consider, consisting of the parameters in the underlying PLM (denoted by $\Theta_{plm}$) and the parameters from our approach (denoted by $\Theta_{ref}$). Note that we will always fix the text encoder from CLIP.

**Fine-tuning for Small PLMs.** For small PLMs, we can perform full-parameter fine-tuning, which updates both $\Theta_{plm}$ and $\Theta_{ref}$. Specifically, given the visually-augmented representations $\mathbf{H}_v$ of VH-words, we directly incorporate them into the embedding layer of the PLM. For a certain VH-word, we insert its visually-augmented representation after the original word embedding, which leverages the visual semantics to enrich the word representations.

**Prompt-tuning for Large-Scale PLMs.** For large-scale PLMs, we fix the parameters in it, *i.e.,* $\Theta_{plm}$ and employ a parameter-efficient prompt-tuning way to optimize it on downstream NLP tasks. Concretely, given the visually-augmented representations $\mathbf{H}_v$ of VH-words, we directly insert them before the *key*, *value* of every layer of PLM, respectively. Then, following the typical prompt-tuning paradigm (Li & Liang, 2021), we only tune the parameters of the reformulation layer (*i.e.,* $\Theta_{ref}$) to control the soft prompts.

Our approach can be generally applied to various PLMs (*e.g.,* BERT (Devlin et al., 2019), BART (Lewis et al., 2020), T5 (Raffel et al., 2020)) and NLP tasks (natural language understanding and text generation). Unlike other complicated visually-augmented methods (Tan & Bansal, 2020; Wang et al., 2022), it is more efficient, without the explicit need of external images or generation model; and meanwhile, it only introduces a relatively small number of parameters (Eq. 2), which are easier to learn.

Table 1: Performance comparison of different methods on NLU tasks, the **BEST** results are highlighted in bold. +*None* denotes that we directly fine-tune the backbone without adding visual information. SBS, VABS, and LBS represent using the syntax-based strategy, visually-enhanced attention based strategy, and learning-based strategy in our approach, respectively. The results of VOKEN and iACE on GLUE are reported from Lu et al. (2022).

| Base Model | Method | SST-2 | QNLI | QQP | MNLI | MRPC | STS-B | Avg. |
|---|---|---|---|---|---|---|---|---|
| CLIP | +None | 73.3 | 74.5 | 72.8 | 68.4 | 74.3 | 73.8 | 72.85 |
| BLIP | +None | 76.3 | 77.4 | 78.8 | 72.5 | 77.8 | 76.4 | 76.53 |
| ALBEF$_{14M}$ | +None | 78.9 | 78.2 | 79.4 | 73.4 | 76.5 | 77.5 | 77.31 |
| BERT$_{base}$ | +None | 89.3 | 87.9 | 87.2 | 79.4 | 81.7 | 84.4 | 84.98 |
| | +VOKEN | 92.2 | 88.6 | 88.6 | 82.6 | 83.5 | 86.0 | 86.83 |
| | +iACE | 91.7 | 88.6 | 89.1 | 82.8 | 85.8 | 86.6 | 87.43 |
| | +VAWI-SBS | **92.9** | 88.4 | 89.6 | 82.2 | 85.5 | 86.9 | 87.58 |
| | +VAWI-VABS | 92.7 | 88.9 | 89.5 | 82.7 | **85.8** | **87.2** | **87.80** |
| | +VAWI-LBS | 92.4 | **89.1** | **89.7** | **83.0** | 85.6 | 86.9 | 87.78 |
| RoBERTa$_{base}$ | +None | 89.2 | 87.5 | 86.2 | 79.0 | 81.4 | 85.4 | 84.78 |
| | +VOKEN | 90.5 | 89.2 | 87.8 | 81.0 | 87.0 | 86.9 | 87.06 |
| | +iACE | 91.6 | 89.1 | **87.9** | 82.6 | 87.7 | 86.9 | 87.63 |
| | +VAWI-SBS | 91.4 | 89.4 | 87.7 | 82.2 | 88.2 | 87.7 | 87.76 |
| | +VAWI-VABS | **91.7** | 89.1 | **87.9** | **82.6** | 88.3 | 88.1 | 87.95 |
| | +VAWI-LBS | 91.6 | **90.6** | **87.9** | 82.4 | **88.5** | **88.3** | **88.21** |

# 4 EXPERIMENTS

## 4.1 EXPERIMENTAL SETUP

**Datesets.** We conduct experiments on four types of NLP tasks. (1) Nature Language Understanding (NLU) task: we extract 6 datasets from the popular GLUE benchmark (Wang et al., 2018); (2) Commonsense reasoning task: we select CommonsenseQA (Talmor et al., 2019), a 5-way multiple choice QA dataset that requires commonsense knowledge; (3) Text generation task: we select CommonGen (Lin et al., 2019b), a constrained text generation task about generative commonsense reasoning. (4) Cross-modal reasoning task: we select SNLI-VE (Xie et al., 2019), to evaluate the capacity of predicting whether the image semantically entails the text.

## 4.2 MAIN EXPERIMENTAL RESULTS

In this part, we conduct a series of experiments on NLU, commonsense reasoning, text generation, and cross-modal commonsense reasoning tasks.

**Evaluation on NLU Tasks.** We present the experimental results of different methods on 6 NLU tasks in Table 1. First, we observe that VL-PTMs perform worse than PLMs, a possible reason is that they have been continually pre-trained on large-scale image-text pairs, which may cause the catastrophic forgetting problem. Second, VaLMs (*i.e.,* VOKEN, iACE ,and VAWI) achieve better performance over PLMs. As VaLMs infuse external visual knowledge into the PLMs, they can help the PLMs better understand the background knowledge of some words (*e.g.,* color, shape, and size of objects). Between the two VaLM baselines, iACE is slightly better. This is because iACE is enhanced based on VOKEN and incorporates an image generation model, so it produces more visual information to utilize. However, the generated images inevitably contain noise and redundant information, which limits the performance gain of iACE.

Finally, by comparing our approach with all baselines, it is obvious that VAWI performs consistently better than them on the six datasets. In our approach, we adopt an efficient and effective way that augments the visually-augmented representations using the text encoder of CLIP to encode the VH-words from the input text. Benefiting from pre-training on large-scale image-text pairs, the text encoder of CLIP has been well aligned with the semantic space of images, so that it can generate high-quality visually-augmented representations of the VH-words to enrich them. Such a way not

only saves the costs of time and computation but also reduces the influence of inevitable noise from retrieved or generated images. It is worth noting that our approach only needs to add 5.09M learnable parameters to the RoBERTa-base, far less than the 192.37M learnable parameters of iACE, but achieves better performance. Additionally, among three VH-words extraction strategies, LBS slightly outperforms others in most NLU tasks. The reason is that LBS incorporates a learnable model-based strategy to select the VH-words. Such a way can adaptively extract proper VH-words with the consideration of the intrinsic knowledge of the PLMs. However, LBS will increase the computation cost due to its involved learnable VH-words extractor layer. Therefore, for efficiency, in the following experiments, we utilize the SBS strategy in our approach for comparison.

**Evaluation on Commonsense Reasoning Tasks.** Following existing works (Lin et al., 2019a), we also rely on a rule-based strategy to extract the examples containing visible objects, to construct a new dataset called CommonsenseQA-3K. It consists of 2,903 and 341 examples in the training set and dev set, respectively. In this way, we can ensure that most of the instances in this dataset require the visual information to enrich the commonsense knowledge. Based on the CommonsenseQA and CommonsenseQA-3k, we also report the results with different amounts of training data, to further evaluate the performance of different methods in the few-shot setting.

We show the experimental results of different methods on the commonsense reasoning tasks in Table 2. We can also see that with the help of the visual information from either retrieved images or our VAWI-SBS, the performance of PLMs can be improved significantly. It indicates that the visual information is indeed helpful to improve PLMs for understanding commonsense knowledge. Besides, our approach outperforms the method using retrieved images from search engines. Our approach omits the image retrieval process due to its inevitably involved noise, and relies on the text encoder of CLIP to augment the visual representations. Such a way can guarantee the relevance between the augmented visual knowledge and the text input, reducing the influence of retrieved noisy images and redundant information. Furthermore, we also perform parameter-efficient tuning on T5-3B-encoder with our approach and boost its performance. It shows that our approach is able to be applied to large-scale PLMs to meet their thirst for visual information.

Table 2: Performance comparison on CommonsenseQA-3k and CommonsenseQA with different amounts of training data. We report the average performance on the dev set over three runs, and the **BEST** results are highlighted in bold. *+Images* denotes that we add retrieved images about the VH-words using web search engines, and encode them via CLIP-ViT.

| Base Model | Method | CommonsenseQA-3k | | | | CommonsenseQA | | | |
|---|---|---|---|---|---|---|---|---|---|
| | | 5% | 10% | 20% | 100% | 5% | 10% | 20% | 100% |
| RoBERTa$_{base}$ | +None | 41.88 | 46.04 | 50.58 | 61.88 | 44.88 | 50.04 | 57.08 | 67.90 |
| | +Images | 42.37 | 48.09 | 52.81 | 64.22 | 45.72 | 51.17 | 58.96 | 69.64 |
| | +VAWI-SBS | **42.94** | **49.27** | **53.97** | **65.10** | **46.51** | **52.44** | **59.87** | **71.01** |
| RoBERTa$_{large}$ | +None | 48.39 | 56.30 | 59.06 | 74.19 | 51.24 | 59.95 | 65.52 | 76.65 |
| | +Images | 49.55 | 57.78 | 61.29 | 75.61 | 52.18 | 60.93 | 66.08 | 78.39 |
| | +VAWI-SBS | **50.27** | **58.17** | **62.22** | **76.54** | **52.98** | **61.97** | **67.40** | **79.19** |
| T5-3B | +None | 70.16 | 73.02 | 75.04 | 81.81 | 71.99 | 75.27 | 77.72 | 82.40 |
| | +Images | 70.96 | 73.60 | 75.91 | 82.40 | 72.87 | 76.17 | 78.71 | 83.64 |
| | VAWI-SBS+PET | **71.52** | **74.19** | **76.49** | **83.61** | **73.58** | **73.58** | **79.66** | **84.56** |

**Evaluation on the Text Generation Task.** As shown in previous experiments, it is useful to improve the performance of VAWI on commonsense reasoning and nature language understanding tasks. Here, we would like to study the effectiveness of our approach on the text generation task (*i.e.,* CommonGen) using large PLMs. As shown in Table 3, our model VAWI also consistently boosts the performance of BART-Large and T5-3b among all metrics. It further shows that our approach can also improve PLMs on the text generation task. As a comparison, we can see that the retrieved images are not very helpful and even cause performance degradation. The reason may be that the text generation task is more sensitive to the inevitable noise from the retrieved images. Finally, the parameter-efficient tuning strategy of our approach also achieves comparable performance with the full-parameter tuning. It indicates that our parameter-efficient strategy is able to efficiently optimize

Table 3: Performance comparison on CommonGen. We also show the performance of parameter-efficient tuning of our approach, denoted as *+PET*. The **BEST** results are highlighted in bold.

| Method | Base Model | BLUE-3 | BLUE-4 | METOR | Rouge-L | CIDER | SPICE |
|---|---|---|---|---|---|---|---|
| BART-large | +None | 42.80 | 32.42 | 31.36 | 57.57 | 16.56 | 32.94 |
|  | +Images | 42.67 | 32.67 | 32.12 | 57.46 | 16.78 | 32.81 |
|  | +VAWI-SBS | **44.56** | **34.17** | **32.47** | **58.46** | **17.23** | **33.67** |
|  | +VAWI-SBS+PET | 43.12 | 33.76 | 32.20 | 58.12 | 16.91 | 33.17 |
| T5-3b | +None | 45.92 | 35.92 | 33.02 | 58.57 | 17.71 | 33.51 |
|  | +Images | 45.69 | 35.50 | 33.55 | 58.94 | 17.51 | 32.91 |
|  | +VAWI-SBS | **47.67** | **37.54** | 33.41 | **59.94** | **18.34** | **34.67** |
|  | +VAWI-SBS+PET | 47.40 | 37.36 | **33.71** | 59.78 | 18.18 | 34.17 |

the parameters of large-scale PLMs, and shows a promising future to apply our approach to much larger PLMs, *e.g.,* GPT-3.

**Evaluation on the Cross-modal Commonsense Reasoning Task.** To verify the generality of our method, we further evaluate the effectiveness of our VAWI on improving a VL-PTM (*i.e.,* ALBEF Li et al. (2021)), and conduct experiments on a cross-modal reasoning dataset, SNLI-VE. Concretely, we implement our approach on ALBEF by inserting the visually-augmented representations after the VH-words embeddings of the text encoder before the multimodal encoder, and keeping others unchanged. As shown in Table 4, our VAWI can also improve the performance of ALBEF using different amounts of training data. It further shows the generality of our approach in VL-PTMs, as it can also provide rich information to enhance the text encoder of VL-PTM, helping it better perform cross-modal reasoning.

Table 4: Results on the test set of SNLI-VE task. The **BEST** results are highlighted in bold.

| Method | SNLI-VE | | | |
|---|---|---|---|---|
|  | 10% | 20% | 50% | 100% |
| ALBEF | 65.46 | 67.52 | 75.47 | 80.91 |
| ALBEF+VAWI +SBS | **65.94** | **68.23** | **76.14** | **81.64** |

## 5 CONCLUSION

In this paper, we proposed a general visually-augmented fine-tuning approach that can be applied to a variety of PLMs and NLP tasks, without using any retrieved or generated images, namely **VAWI**. Specifically, we first identified and extracted the visually-hungry words (VH-words) from input text via a token selector, where three different methods have been proposed, including syntax-, attention- and learning-based strategies. Then, we adopted a fixed CLIP text encoder to generate the visually-augmented representations of these VH-words. As it has been pre-trained by visual-language alignment task on large-scale corpus, it is capable of injecting visual semantics into the aligned text representations. Finally, we transformed the visually-augmented features into several pre-designed visual prompts based on VH-words, and inserted them into PLMs to enrich the visual semantics of word representations in PLMs. Extensive experiments conducted on 10 NLP tasks, *i.e.,* GLUE benchmark, CommonsenseQA, CommonGen, and SNLI-VE show that our approach can consistently improve the performance of BERT, RoBERTa, BART, and T5 at different scales, and outperform several competitive baselines significantly. Besides, the visual prompts of our framework can also be used for parameter-efficient tuning, which boosts the performance of T5-3b.

In the future, we will investigate more effective approaches to extract visual-hungry words from the input text, and incorporate more effective VL-PTMs to augment the visual representations. Besides, we will also devise more effective visually-augmented pre-training strategies without images, to further improve the performance of PLMs.

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

## A    EXPERIMENTAL SETUP

**Implementation Details.** We implement all baselines and our method based on Huggingface Transformers (Wolf et al., 2020). For all baselines, we set their hyper-parameters according to the original papers. In our approach, we leverage the text encoder of CLIP (ViT-B/32) to implement the learnable model-based VH-words extractor and generate the visual representations of VH-words in the visual knowledge augmentation module. The hidden size of visual representations is set to 512. For different NLP tasks, we tune the number of visually hungry words in {2, 3, 4, 5}. During fine-tuning, we perform parameter-efficient tuning on T5-3b and BART-Large, and full-parameter tuning on other PLMs. For all tasks and all backbones, we utilize Adam as the optimizer, set the learning rate to 2e-5, weight decay to 0.01, and a linear warmup for the first 6% steps. For GLUE, GommonGen, and SNLI-VE datasets, we fine-tune our model for 3 epochs with a batch size of 32. For CommonsenseQA, we tune our model for 10 epochs with a batch size of 32. We use the cross-entropy loss for classification and mean squared error loss for regression. All experiments are conducted on a V100 GPU.

**Baseline Models.** We compare our approach with the following baselines, including pre-trained language models (PLMs), visual-language pre-trained models (VL-PTMs), and visually-augmented pre-trained language modes (VaLMs). (1) **PLMs**: We choose BERT (Devlin et al., 2019), RoBERTa (Liu et al., 2019), BART (Lewis et al., 2020), T5 (Raffel et al., 2020) as the PLM backbones, and directly fine-tune them as baselines. (2) **VL-PTMs**: We select ALBEF (Li et al., 2021), BLIP (Li et al., 2022), and CLIP (Radford et al., 2021) as baselines, which have been pre-trained on large-scale image-text pairs via cross-modal contrastive learning. (3) **VaLMs**: we select VO-KEN (Tan & Bansal, 2020) and iACE (Lu et al., 2022), which introduce the visual information into PLMs by pre-training on retrieved images and fine-tuning on generated images, respectively.

## B    ABLATION STUDY

### B.1    ABLATION STUDY ON VISUAL KNOWLEDGE AUGMENTATION

In this part, we conduct a series of experiments to verify whether the improvement of our approach derives from the augmented visual knowledge about the VH-words.

**The Effect of the Source of Visual Representations.** We first propose three variants that incorporate powerful PLMs, *i.e.,* RoBERTa-base, T5-Large, and T5-3b respectively, to replace the text encoder of CLIP in our framework. We also replace the generated visual representations from the text encoder of CLIP with random noise, to investigate the importance of the visual representations. As shown in Table 5, we can see that our approach is better than all the variants, even T5-3b with billion-scale parameters. It indicates that CLIP-base is more effective to augment visual knowledge to improve the performance of PLMs. Besides, our approach also outperforms the variant using random noise as the visual representation, showing the worse performance among all the variants. It also shows the importance of visual representations, as they indeed contain the visual knowledge that the PLM is hungry for.

Table 5: Performance comparison of different sources of visual representation in our approach. The base model is RoBERTa-base.

| Source of visual representation (Params) | CSQA-3k | CSQA | SST-2 | QQP | STS-B | QNLI |
|---|---|---|---|---|---|---|
| Random Noise (0M) | 61.59 | 66.78 | 89.13 | 86.27 | 85.13 | 87.22 |
| RoBERTa-large (355M) | 61.18 | 67.17 | 89.43 | 86.53 | 85.60 | 87.77 |
| T5-large-encoder (375M) | 62.21 | 67.87 | 89.71 | 86.67 | 86.40 | 87.94 |
| T5-3b-encoder (1500M) | 63.10 | 68.42 | 90.24 | 86.96 | 86.93 | 88.21 |
| CLIP-base (52M) | **65.10** | **71.07** | **91.41** | **87.72** | **87.67** | **89.40** |

**The Effect of the Stronger VL-PTMs.** In our work, we choose CLIP-base to enhance PLMs, as it has been pre-trained on a large-scale image-text dataset. Generally, a stronger VL-PTM would

be more promising to further improve the performance. Here, we replace our CLIP-base model with some stronger VL-PTMs, *e.g.,* ALBEF (Li et al., 2021), UniCL-base (Yang et al., 2022), and CLIP-large. Concretely, ALBEF leverages more pre-training tasks (*e.g.,* MLM, ITM, and ITC), UniCL utilizes more high-quality pre-training data, and CLIP-large increases the scale of model parameters. We evaluate the above variations on CSQA-3k, QQP, and SST-2, and the results are shown in Table 6. We can see that UniCL and CLIP-large outperform CLIP-base. It indicates that the VL-PTMs with larger scale of model parameters or more high-quality pre-training data are more capable of augmenting useful visual knowledge for PLMs. Considering the efficiency, CLIP-base is also a good choice in our approach, and we will investigate more proper VL-PTMs in the future.

Table 6: Performance comparison of visual representations from different VL-PTMs in our approach. The base model is RoBERTa-base.

| The text encoder of different VL-PTMs (Params) | CSQA-3k | SST-2 | QQP |
|---|---|---|---|
| Random Noise (0M) | 61.59 | 89.23 | 86.21 |
| ALBEF (110M) | 63.34 | 90.72 | 87.17 |
| CLIP-base (52M) | 65.10 | 91.41 | 87.72 |
| UniCL-base (52M) | 65.98 | 91.75 | 88.07 |
| CLIP-large (123M) | **66.27** | **92.10** | **88.31** |

**The Effect of the Pre-trained Dataset of VL-PTMs.** We notice that the pre-training dataset of VL-PTMs is different from PLMs. Here, we investigate whether the captions or images from the large-scale image-text pairs contribute more to the performance gain of our approach. To verify it, we pre-train a new PLM only using the captions data. Following the setting of ALBEF, we utilize the pre-trained parameters of BERT to initialize this model and only extract the captions from the pre-training data of ALBEF (totally 14.5M sentences). After pre-training on these captions until convergence, we utilize this model to replace CLIP-base in our approach and keep other settings unchanged. We conduct experiments on commonsense reasoning and NLU tasks to evaluate its effectiveness for augmenting visual knowledge. As shown in Table 7, we can see that such a variation underperforms ALBEF and our approach, and even leads to performance degradation on CSQA task. It indicates that during pre-training the image data is an important resource for learning visual knowledge in VL-PTMs. Only text data (*i.e.,* captions) can not provide sufficient visual knowledge that PLMs are hungry for. Therefore, after pre-learned on large-scale text-image pairs, CLIP can absorb the useful visual knowledge from the images and inject them into PLMs in our approach. It further indicates that the improvement of our method is due to the involvement of the visual information about the VH-words.

Table 7: Performance comparison of visual representations pre-trained using different pre-training data in our approach. The base model is RoBERTa-base.

| The text encoder of different VL-PTMs (Params) | CSQA-3k | CSQA | SST-2 | STS-B | MNLI |
|---|---|---|---|---|---|
| None | 61.59 | 67.90 | 89.23 | 85.46 | 79.06 |
| BERT pre-trained on captions (110M) | 62.17 | 67.56 | 89.58 | 85.73 | 79.24 |
| ALBEF (110M) | 63.64 | 68.47 | 90.72 | 87.17 | 80.86 |
| CLIP-base (52M) | **65.10** | **71.07** | **91.41** | **87.73** | **82.27** |

## B.2 ABLATION STUDY ON VISUALLY-ENHANCED FINE-TUNING

**Different Insertion Positions of Visual Representations.** In our visually-enhanced fine-tuning framework, we insert the visual representation of the VH-word after its original word embedding. To verify its effectiveness, we propose three variants of it that do not insert, insert all visual representations of VH-words before and after the input text, respectively. As shown in Table 8, we can observe that all these variants would lead to a performance decrease. It demonstrates that a proper position to insert the visual representation is important for the utilization of augmented visual representations. By inserting them after the word embeddings of corresponding VH-words, PLMs can

effectively aggregate the visual representations to enrich the word representations, leading to better performance on downstream NLP tasks.

Table 8: Performance comparison w.r.t. different insertion positions of visual representations. The base model is RoBERTa-base.

| Insert Positions | CSQA-3k | | | |
|---|---|---|---|---|
| | 5% | 10% | 20% | 100% |
| Not insert | 41.88 | 46.04 | 50.58 | 61.88 |
| Before input text | - | 39.77 | 44.86 | 57.47 |
| After input text | - | 40.23 | 45.67 | 58.08 |
| After the VH-words | **42.94** | **49.27** | **53.97** | **65.10** |

## C  FURTHER ANALYSIS

**The Number of VH-words.**  Our approach has an important hyper-parameter required to tune, such as the number of VH-words. VH-words can supply visual knowledge that PLMs may be hungry for. Here, we would like to study whether more VH-words are better to improve performance. We conduct experiments on the QQP and CSQA-3K datasets using RoBERTa-base as the backbone, and present the results in Figure 2. We can see that with the increase of the number of VH-words, the performance gain of our approach first increases and then decreases. A possible reason is that too many VH-words may also introduce noisy or redundant information (*e.g.,* not very relevant words), which would also influence the fine-tuning performance. Instead, it is also more efficient to select a few VH-words (*e.g.,* two words for CSQA-3k) for deploying our approach in large-scale PLMs.

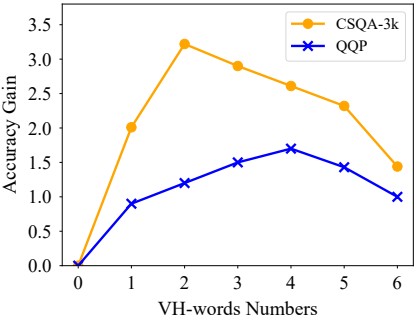

Figure 2:   Performance comparison w.r.t. different numbers of VH-words.

**The Computation Latency of the Proposed Methods.** In our VAWI, we fix the model parameters of CLIP-base to preserve the visual knowledge. Such a way can also decrease the computation costs during training and inference. To verify it, we report the mean training and inference latency per batch on the CSQA-3k dataset of our method and baselines on RTX3090 GPU, where all these methods utilize RoBERTa-base as the backbone. As shown in the Table 9, we can see that our proposed VAWI-SBS and VAWI-ABS would not increase the latency too much. For VAWI-LBS, as it requires a PLM and a VL-PTM to adaptively select the VH-words, it will relatively increase the much latency. As shown in Table 1, we can see that all the three variants achieve comparable performance in 6 NLU datasets. Therefore, it is more efficient and effective to select the SBS and ABS variations in our approach. Despite it, we can see that all our variants own less latency than iACE, since our approach does not require a time-consuming image generation process. And as shown in Table 1, our approach can also achieve better performance.

Table 9: The computation latency during training and inference.

| Computation Costs | RoBERTa-base | +VAWI-SBS | +VAWI-ABS | +VAWI-LBS | +Voken | +iACE |
|---|---|---|---|---|---|---|
| Training Time (s) | 0.506 | 0.587 | 0.680 | 0.893 | 0.506 | 1.138 |
| Inference Time (s) | 0.182 | 0.241 | 0.308 | 0.486 | 0.182 | 0.512 |

**The Effect of the Improper Visually-hungry Words.** To analyze how the quality of the VH-words affects the performance of our approach, we further conduct the experiments on CSQA-3K and two NLU tasks SST-2 and QQP from GLUE, to show the effect of insufficient VH-words on our model performance. After extracting the VH-words, we remove part of them and only randomly

sample 0%, 20%, and 50% VH-words for augmentation. As shown in Table 10, we can see that with the decreasing of the sampling probability, the performance of our approach degrades gradually. It indicates that not enough VH-words would degrade the performance of our approach.

Table 10: The effect of the improper visually-hungry words. The base model is RoBERTa-base.

| The proportion of correct visual words | CSQA-3k | SST-2 | QQP |
|---|---|---|---|
| 0 % | 61.60 | 89.57 | 87.63 |
| 20 % | 62.17 | 89.44 | 87.40 |
| 50 % | 64.22 | 91.73 | 89.20 |
| 100 % | **65.10** | **92.93** | **89.74** |
| None | 61.88 | 89.23 | 86.21 |

**The Interpretability of Augmented Embeddings.** In this part, we show how our augmented embeddings infuse visual knowledge into the PLM. Concretely, we show the attention distributions of a PLM (*i.e.,* RoBERTa-base) in the last few layers before and after infusing visually-augmented representations on CSQA. As shown in Table 11, we can see that the [CLS] tokens pay more attention to the visually-hungry words (VH-words) and their visually-augmented representations, and the VH-words also pay more attention to their visually-augmented representations. It shows that the injected visually-augmented representations indeed provide useful knowledge, which guides the PLM to focus on more important tokens and also improves the representations of the VH-words and the [CLS] token.

**Case Study of Extracted Visually-hungry Words.** In this part, we show the visually-hungry words (VH-words) extracted by syntax-, attention- and learning-based strategies in the Table 12, Table 13, Table 14 and Table 15. We can see that the three strategies would extract slightly different VH-words. The reason is that the three strategies are based on different techniques to identify the VH-words. As we can see, the cases show that most of the extracted VH-words by our strategies are generally related to some visual semantics, *e.g.,* spider, two eyes. Although such VH-words can not perfectly cover all the visual semantics, they actually contain most of the important words that the PLMs may be hungry for, *e.g.,* red and yellow. Besides, we can also see that the VH-words extracted by our three strategies may not perfectly align with human judgment. In fact, it is also hard for humans to determine proper rules to identify VH-words, *e.g.,* people, human, and water. In addition, as the learned knowledge of PLM is a black box, it is also difficult for humans to judge the usefulness of our extracted VH-words for PLMs.

Table 11: The attention maps of the self-attention layers on RoBERTa-base and our approach.

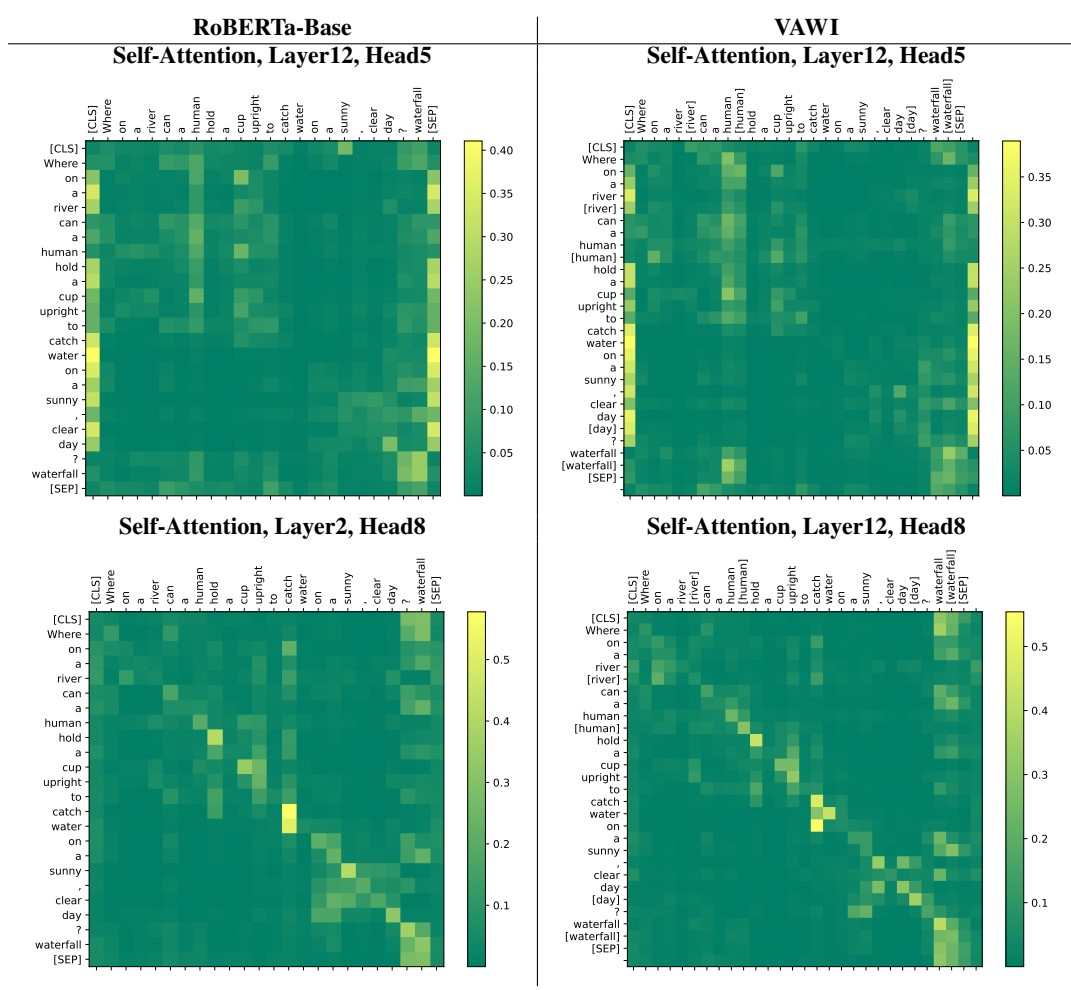

Table 12: The first instance from the CommonsenseQA dataset. The extracted visually-hungry words are highlighted in green.

| Input |
|---|
| *Input sentence:* Unlike a spider and his many sight seers, people only have what? two eyes. |

| **Syntax-based Strategy** |
|---|
| Unlike a spider and his many sight seers, people only have what? two eyes |

| **Attention-based Strategy** |
|---|
| Unlike a spider and his many sight seers, people only have what? two eyes. |

| **Attention-based Strategy** |
|---|
| Unlike a spider and his many sight seers, people only have what? two eyes. |

Table 13: The second instance from the CommonsenseQA dataset. The extracted visually-hungry words are highlighted in green.

| |
|---|
| **Input** |
| *Input sentence:* Where on a river can a human hold a cup upright to catch water on a sunny, clear day? waterfall. |
| **Syntax-based Strategy** |
| Where on a river can a human hold a cup upright to catch water on a sunny, clear day? waterfall. |
| **Attention-based Strategy** |
| Where on a river can a human hold a cup upright to catch water on a sunny, clear day? waterfall. |
| **Attention-based Strategy** |
| Where on a river can a human hold a cup upright to catch water on a sunny, clear day? waterfall. |

Table 14: The instance from the SST-2 dataset. The extracted visually-hungry words are highlighted in green.

| |
|---|
| **Input** |
| *Input sentence:* the mesmerizing performances of the leads keep the film grounded and keep the audience riveted. |
| **Syntax-based Strategy** |
| the mesmerizing performances of the leads keep the film grounded and keep the audience riveted. |
| **Attention-based Strategy** |
| the mesmerizing performances of the leads keep the film grounded and keep the audience riveted. |
| **Attention-based Strategy** |
| the mesmerizing performances of the leads keep the film grounded and keep the audience riveted. |

Table 15: The instance from the QQP dataset. The extracted visually-hungry words are highlighted in green.

| |
|---|
| **Input** |
| *Input sentence:* How do I sell dry Moringa leaves powder in Indian market? Can I use the moringa leaves that are already starting to turn yellow or yellowish? |
| **Syntax-based Strategy** |
| How do I sell dry Moringa leaves powder in Indian market? Can I use the moringa leaves that are already starting to turn yellow or yellowish? |
| **Attention-based Strategy** |
| How do I sell dry Moringa leaves powder in Indian market? Can I use the moringa leaves that are already starting to turn yellow or yellowish? |
| **Attention-based Strategy** |
| How do I sell dry Moringa leaves powder in Indian market? Can I use the moringa leaves that are already starting to turn yellow or yellowish? |

