# OpenReview forum: "Visually-augmented pretrained language models for NLP Tasks without Images"
_ICLR.cc/2023/Conference — Submitted to ICLR 2023_

### Official Review · Reviewer_TuCV · 2022-10-18

**Confidence:** 3
**Correctness:** 2
**Technical Novelty And Significance:** 3
**Empirical Novelty And Significance:** 3
**Recommendation:** 5

**Clarity, Quality, Novelty And Reproducibility:**

Selective incorporation of CLIP features into a text-only model is an interesting idea that I haven't seen before. The experiments are extensive and straightforward to understand.

**Strength And Weaknesses:**

The question the authors address is one of the "holy grails" of
vision-and-language research --- it seems like visual information
should be able to augment a text only model's capacity (sometimes
called "vision-for-language"). Yet, the results to date haven't been
definitive. Here, the authors results suggest that
the visually grounded textual representations of CLIP can do just that.
This method outperforms Vokenization and iACE, which are two prior
visual knowledge transfer to text-only tasks works, which is
impressive.

My biggest concern is that, despite the presented experiments, I am
not convinced that visual information is the reason why performance
improves. The authors presented a compelling case that CLIP textual
features uniquely improve performance vs. text-only alternatives. But
--- does this really mean that visual knowledge transfer is occurring?
For example, what visual knowledge is required for SST-2, i.e., which
instances does the model improve performance over, and are those one's
extra "visual"? The authors allude to color/shape/size information
being what is transferred, but the experiments do not complete that
argument.

It's a very difficult argument to make, and I commend the authors for
their ambition in providing additional results in this sometimes
difficult-to-analyze domain. I appreciate the ablations the authors
ran in the appendix. And yet, despite these efforts, the core
conclusion of the paper requires that visual knowledge is why CLIP
text representations do better --- I am still somewhat unconvinced.

Alternate explanations that are not possible to rule out include:

- CLIP's pretraining dataset ImageWebText, contains different text
  compared to the usual language-only models. Maybe the text-only
  information there is why they help? Prior work on commongen suggests
  incorporation of additional textual information can improve performance:

@inproceedings{wang-etal-2021-retrieval-enhanced,
    title = "Retrieval Enhanced Model for Commonsense Generation",
    author = "Wang, Han  and
      Liu, Yang  and
      Zhu, Chenguang  and
      Shou, Linjun  and
      Gong, Ming  and
      Xu, Yichong  and
      Zeng, Michael",
    booktitle = "Findings of the Association for Computational Linguistics: ACL-IJCNLP 2021",
    month = aug,
    year = "2021",
    address = "Online",
    publisher = "Association for Computational Linguistics",
    url = "https://aclanthology.org/2021.findings-acl.269",
    doi = "10.18653/v1/2021.findings-acl.269",
    pages = "3056--3062",
}

- CLIP's contrastive objective at the level of sentences is different
  enough from language modeling objectives such that any contrastive
  model's features might add predictive accuracy.

- The authors ran a large number of permutations of their model ---
  could it be that the performance gains of a few accuracy points are
  due mostly in part to better hyperparameter optimization?

While definitively making the visual knowledge transfer case is
perhaps beyond the scope of this work --- I nonetheless could envision
a number of missing additional experiments in this setup:

- Which words does the model select as "visually hungry"? Do they
  align with the syntax-based method? Do they align with human judgments
  of visual-ness, e.g., from:

Douglas L. Nelson, Cathy L. McEvoy, and Thomas A.
Schreiber. 2004. The University of South Florida free association,
rhyme, and word fragment norms.  Behavior Research Methods,
Instruments, & Computers, 36(3):402–407.

- Which instances across the NLP tasks does performance improve most
  for? Are these cases where we would expect visual information to
  help?

**Summary Of The Paper:**

The authors demonstrate that performance on
GLUE/Commongen/CommonsenseQA can be improved by selectively
incorporating textual CLIP representations. This method outperforms
prior vision-for-language methods like Vokenization. Ablations in the
appendix substituting random/T5/etc. features for the CLIP features
suggest that this effect is not due to architecture choices, but
something unique about the CLIP textual features.

**Summary Of The Review:**


Overall --- the authors make some promising experimental steps towards
an exciting goal: showing that "visual knowledge" can be transferred
to text-only models with performance improvements for downstream
tasks.  However, the argument is incomplete: there are plausible
alternative hypotheses that could explain CLIP features helping
language models beyond "visual knowledge:" and, some experiments that
could quantify additional aspects of the potential visual knowledge
transfer are missing.

=== after response:

The new experiments the authors ran to address my core concern are interesting initial steps, and I raised my score accordingly. However, I am still not 100% convinced that visual information is the reason why CLIP base features are better. 400M image captions is quite a lot of novel data, which might be the bottleneck in the scaling regime the authors consider in their additional creative experiment (first one described in response to YcbG). It's a great initial type of experiment to run, but, because the authors are quite ambitious in their work, the extraordinary claim of visual knowledge transfer, for me, requires more evidence than just this initial result on ALBEF.

---

> ### Author Response · Authors · 2022-11-19
> **Response to the concerns about the reason for performance gains.**
>
> We sincerely thank the reviewer for the insightful suggestions. We have carefully revised our manuscript and listed the detailed responses to the concerns in the reviewer's suggestions.
>
> **Q1.The reason for the improvement of our proposed method.**
>
> In our approach, the performance improvement derives from the augmentation and utilization of the visual knowledge, where the visual encoder (i.e., CLIP-base) and the proposed visually-hungry words selection strategies are very important. To verify the contribution of the two components, we added corresponding ablation and variation studies. First, as shown in Table 5 of our manuscript and the response to Q1(a) of reviewer YcbG, we replace our CLIP-base with other PLMs, and see that the performance degrades. It indicates the effectiveness of CLIP-base, as it has been pre-trained with large-scale image-text datasets via cross-modal contrastive learning. For the proposed VH-words selection strategies, as shown in the response to Q1 and Q5 of A4xX, we can see that reducing the number of VH-words or selecting bad VH-words would also degrade the performance of our method. It also demonstrates that a proper VH-words selection strategy is essential to our approach, and our proposed SBS, VABS, and LBS are effective ways.
>
> **Q2. I am not convinced that visual information is the reason why performance improves.**
>
> As shown in Table 5 of our manuscript and the response to Q1(a) of reviewer YcbG, we replace our CLIP-base with other PLMs, and see that the performance degrades. It indicates the effectiveness of CLIP-base, as it has been pre-trained with large-scale image-text datasets via cross-modal contrastive learning. To further investigate if the pre-trained image-text dataset is essential, in the response to Q1(b) of reviewer YcbG, we can see that VL-PTM generally performs better than single-modal PLMs, and more powerful VL-PTMs using larger-scale model parameters or more high-quality image-text pairs are more effective. It further indicates that the learned visual information of the VL-PTMs is important to our effectiveness. Besides, in the response to Q1(c) of reviewer YcbG, only using the captions can not achieve good performance as our approach. Such a result further indicates that the pre-learned visual information from the VL-PTM is the key to our effectiveness. In summary, our added experiments have revealed that the major contribution of our approach derives from the learned visual information of the VL-PTM. Once removing or replacing the visual information by others (e.g., text-only captions or large PLMs), the performance of our approach will degrade.
>
> **Q3. Are the performance gains due to better hyper-parameter optimization?**
>
> In our experiments, to guarantee a fair comparison, we do not perform any parameter tuning. We just use the same hyper-parameters as the baselines for all tasks. Concretely, we utilize Adam as the optimizer, set the learning rate to 2e-5, weight decay to 0.01, and a linear warmup for the first 6% steps. For GLUE, GommonGen, and SNLI-VE datasets, we fine-tune our model for 3 epochs with a batch size of 32. For CommonsenseQA, we tune our model for 10 epochs with a batch size of 32.

---

> > ### Comment · Reviewer_TuCV · 2022-11-29
> > **response acknowledged**
> >
> > Thanks for taking the time to consider my review and to respond to the points I made. Accordingly, I will update my review score because of these new experiments/steps taken to address the concern of explaining the true reason for the performance gain. Not all questions are answered fully, but these are promising additional steps.

---

> > > ### Author Response · Authors · 2022-12-06
> > > **The new response to the remaining concerns of the reviewer.**
> > >
> > > **We sincerely thank the reviewer for the updated comments and score. And we list our new response to the reviewer’s concerns with experimental results and analyses as follows.**
> > >
> > > Here, we systematically reorganize the response to the main concern of the reviewer, whether the improvement of our approach derives from the augmented visual information of CLIP.  Compared to PLMs, CLIP has two main differences. Concretely, CLIP’s pre-trained corpus consists of large-scale image-text pairs, and it relies on the contrastive learning objective for pre-training. Therefore, we further conduct variation study experiments from these two perspectives and the implementation details and analyses are as follows:
> > >
> > > **a.The effect of image data in the pre-training corpus of the vision-language pre-trained models (VL-PTMs).**
> > > We have investigated whether the image or captions from the large-scale image-text pairs contribute more to the performance gain of our approach in our manuscript and the response to the reviewer YcbG. The results are as follows. We can see that only using the captions can not achieve good performance as our approach. Such a result further indicates that image data during pre-training is an important resource for learning visual knowledge in VL-PTMs, and only text data (i.e., captions) can not provide sufficient visual knowledge that PLMs are hungry for.
> > >
> > > \\begin{array} {|c|c|c|c|}
> > > \\hline
> > >  \\textrm{The text encoder of different VL-PTMs~(Params)} & \\textrm{CSQA-3k} & \\textrm{CSQA}         & \\textrm{SST-2}  & \\textrm{STS-B}  & \\textrm{MNLI} \\\\
> > > \\hline
> > > \\textrm{Noise \(0M\)}  & 61.59            & 67.90                &  89.23           & 85.46          & 79.06 \\\\
> > > \\hline
> > > \\textrm{BERT pre-trained on captions \(110M\)}& 62.17           &  67.56                &  89.58       & 85.73          & 79.24   \\\\
> > > \\hline
> > > \\textrm{ALBEF \(110M\)}                 & 63.64            & 68.47               &  90.72           & 87.17          & 80.86 \\\\
> > > \\hline
> > > \\textrm{CLIP-base \(52M\)}              & \textbf{65.10}   &  \textbf{71.07}       &  \textbf{91.41}  & \textbf{87.73} & \textbf{82.27}
> > > \\end{array}
> > >
> > > **b.The effect of pre-training using contrastive learning objective.**
> > > As the reviewer mentioned, the CLIP's contrastive objective is different from the PLMs’ objective, e.g., mask language model and language model. The difference may enhance the applicability of CLIP as an information resource for augmented representations and improve the performance of PLMs. Here, we investigate whether the contrastive objective contributes to the performance gain of our approach. Concretely, we utilize the pre-trained parameters of BERT to initial a model. Then, we pre-train this model on the captions of ALBEF with the contrastive objective of SimCSE. After pre-training until convergence, we utilize this model to replace CLIP-base in our approach and keep other settings unchanged. We conduct experiments on commonsense reasoning and NLU tasks, and the results are shown as follows. As we can see, the variation underperforms ALBEF and our approach and even leads to performance degradation on the CSQA-3K, CSQA, and MNLI datasets. It further indicates that image data during pre-training is essential for learning visual knowledge in VL-PTMs, and the performance gains of our approach are not due to the contrastive objective at the level of sentences.
> > >
> > > \\begin{array} {|c|c|c|c|c|c|}
> > > \\hline
> > >  \\textrm{The text encoder of different VL-PTMs~(Params)} & \\textrm{CSQA-3k} & \\textrm{CSQA}         & \\textrm{SST-2}  & \\textrm{STS-B}  & \\textrm{MNLI} \\\\
> > > \\hline
> > > \\textrm{None \(0M\)}  &61.88	 &67.90	&89.22	&85.40	&79.07 \\\\
> > > \\hline
> > > \\textrm{BERT pre-trained on captions with contrastive learning \(110M\)} &61.58	&67.23	&89.45	&85.93	&78.69   \\\\
> > > \\hline
> > > \\textrm{ALBEF \(110M\)}                 & 63.64            & 68.47               &  90.72           & 87.17          & 80.86 \\\\
> > > \\hline
> > > \\textrm{CLIP-base \(52M\)}              & \textbf{65.10}   &  \textbf{71.07}       &  \textbf{91.41}  & \textbf{87.73} & \textbf{82.27}
> > > \\end{array}

---

> > > > ### Comment · Reviewer_TuCV · 2022-12-12
> > > > **response once again acknowledged**
> > > >
> > > > Hi authors --- thanks for your extensive additional experiments, and your additional reminder pings. I'm doing my best to keep up. These are great additional things to include in your submission, and I'm happy to see peer review in action. That is, it seems like compared to the initial submission, you've added many, many new results based on our feedback.
> > > >
> > > > A brief summary, which may also be helpful for reviewers/ACs to keep up with the additional content.
> > > >
> > > > 1. Q: is knowledge is derived just from text captions? new experiment (https://openreview.net/forum?id=YWVkyLV53X&noteId=5X8t62gq-2): training BERT on ALBEF corpus of captions and adding ALBEF comparison. CLIP still does better.
> > > >
> > > > 2. Q: is it just the contrastive objective? new experiment (https://openreview.net/forum?id=YWVkyLV53X&noteId=mh4n2eMJKcZ): BERT + constrastive captions. CLIP is still better, and the contrastive objective actually hurts performance.
> > > >
> > > > 3. Q: what if we don't freeze the text representations? new experiment (https://openreview.net/forum?id=YWVkyLV53X&noteId=luoaFxRvJWV): training unfreezing VL-PTM text encoders hurts.
> > > >
> > > > My remaining concerns are about 1/2:
> > > >
> > > > 1. This takes an admirable step, but still leaves open the possibility that the 30X times larger dataset ImageWebText vs. 14.5M ALBEF corpus is causing the gap.
> > > >
> > > > 2. Again, this is an admirable step, but still leaves open the possiblity that a contrastive objective at the level of CLIP (batch size 32K vs. ... actually --- I am not sure how big the batch is here, I couldn't find in comment or updated paper) is a fundamentally different learning regime which requires different representations to address.
> > > >
> > > > ===
> > > >
> > > > I am not trying to be purposefully difficult/nit-picky: I think your initial experiments, significant new experiments, and continued discussion are making great strides towards better addressing the initial question. It's just that the claim of visual knowledge transfer is very, very hard to validate. While initial+continued experiments cannot rule "visual knowledge transfer" out, I still think the argument isn't quite suggestive enough for me to buy:
> > > >
> > > > - The qualitative examples of visually hungry words are nice (https://openreview.net/forum?id=YWVkyLV53X&noteId=1ItsxoWvD7) but... why is "sunny clear day" not visual, but "waterfall" is? There doesn't seem to be a quantitative investigation of if the highlighted words are actually more visually concrete than the others. The experiment I suggested, correlating with human judgements from an additional resource, are not provided --- it's not 100% clear to me the "right" way to conduct this experiment with proper controls, but it would have been appreciated
> > > >
> > > > - I similarly stand by a point from my original review: "... what visual knowledge is required for SST-2, i.e., which instances does the model improve performance over, and are those one's extra "visual"? The authors allude to color/shape/size information being what is transferred, but the experiments do not complete that argument." I think this largely still holds: do performance improvements actually happen on instances that are more "visual"? or are they just random?

---

> > > > > ### Author Response · Authors · 2022-12-13
> > > > > **The new response to the remaining concerns of the reviewer.**
> > > > >
> > > > > **We sincerely thank the reviewer for the careful reading and insightful suggestions. And we list our new response to the reviewer’s concerns with analyses as follows.**
> > > > >
> > > > > **(a). The concern about the pre-trained corpus gap between CLIP and ALBEF.**
> > > > >
> > > > > We agree with the reviewer's suggestion very much. However, a major problem is that the pre-trained corpus of CLIP ImageWebText is not available now. Therefore, we have to select the pre-trained data of ALBEF for substitution. Using the same data from ALBEF, we suggest a more fair comparison between using BERT pre-trained on just the captions and using ALBEF to augment the visual information in our approach. Here, ALBEF has been pre-trained using the same data while including the images. As shown in Table, we can see that using ALBEF can perform better. It shows the effectiveness derived from the images from the pre-trained data.
> > > > >
> > > > > \\begin{array} {|c|c|c|c|}
> > > > > \\hline
> > > > >  \\textrm{The text encoder of different VL-PTMs~(Params)} & \\textrm{CSQA-3k} & \\textrm{CSQA}         & \\textrm{SST-2}  & \\textrm{STS-B}  & \\textrm{MNLI} \\\\
> > > > > \\hline
> > > > > \\textrm{BERT pre-trained on captions \(110M\)}& 62.17           &  67.56                &  89.58       & 85.73          & 79.24   \\\\
> > > > > \\hline
> > > > > \\textrm{ALBEF \(110M\)}                 & 63.64            & 68.47               &  90.72           & 87.17          & 80.86 \\\\
> > > > > \\hline
> > > > > \\end{array}
> > > > >
> > > > > **(b).The concern about the batch size in pre-training with the contrastive objective.**
> > > > >
> > > > > Due to hardware limitations, we cannot use the same batch size as the CLIP in our pre-training. But we notice that both ALBEF and CLIP use the cross-modal contrastive objective in pre-training. Therefore, we have to select the pre-trained data of ALBEF for substitution. For a fair comparison, we use the same data from ALBEF and the contrastive objective to pre-train BERT with a batch size of 512, which follows the pre-trained setting of ALBEF. Here, ALBEF has been pre-trained using the same data while including the images. The results are shown as follows, we can see the variation underperforms our proposed method on ALBEF. It further shows the effectiveness and necessity derived from the images from the pre-trained data.
> > > > >
> > > > > \\begin{array} {|c|c|c|c|c|c|}
> > > > > \\hline
> > > > >  \\textrm{The text encoder of different VL-PTMs~(Params)} & \\textrm{CSQA-3k} & \\textrm{CSQA}         & \\textrm{SST-2}  & \\textrm{STS-B}  & \\textrm{MNLI} \\\\
> > > > > \\hline
> > > > > \\textrm{BERT pre-trained on captions with contrastive learning \(110M\)} &61.58	&67.23	&89.45	&85.93	&78.69   \\\\
> > > > > \\hline
> > > > > \\textrm{ALBEF \(110M\)}                 & 63.64            & 68.47               &  90.72           & 87.17          & 80.86 \\\\
> > > > > \\hline
> > > > > \\end{array}
> > > > >
> > > > > **(c). The concern about the qualitative examples of visually-hungry words.**
> > > > >
> > > > >
> > > > > Different from the syntax-based strategy, visually-enhanced attention- and learning-based strategy calculate scores and select top-K ranked words as visually-hungry words (VH-words). Hence, these two strategies prefer to select VH-words that contain more significant visual information for PLM. As the reviewer mentioned case, the “waterfall” is a key entity to discriminate whether this sentence is consistent with commonsense knowledge. But “sunny, clear day” contains redundant visual information in this case. As we can see, the two strategies filter out “sunny, clear day” and keep “waterfall”. It further indicates that these strategies can filter out irrelevant and redundant visual information to some extent.
> > > > >
> > > > >
> > > > > In fact, we are dedicating to finding the VH-words that are helpful for improving different pre-trained language models (PLMs) and downstream tasks, mostly focusing on the task performance. Actually, the definition of VH-words is hard to determine, there should be different standards for different PLMs and different tasks, even for different researchers. Therefore, instead of struggling to design clear and exclusive rules to detect the VH-words, we would rather give it a slightly ambiguous shape and propose three valuable extraction strategies to inspire other researchers. In this work, our major focus is to reveal that augmenting visual information for improving PLMs may not require external images, but a pre-trained VLP is enough. Our experimental results all prove that such a way can be more effective than existing works using retrieved or generated images in various PLMs, and various tasks in Table 1, Table 2, Table 3, and Table 4 of our manuscript. In future work, we will try to clearly determine what are VH-words, how to effectively select us, and release annotated VH-words datasets for usage.

---

> > > > > ### Author Response · Authors · 2022-12-13
> > > > > **Title: The new response to the remaining concerns of the reviewer**
> > > > >
> > > > > **We sincerely thank the reviewer for the careful reading and insightful suggestions. And we list our new response to the reviewer’s concerns with analyses as follows.**
> > > > >
> > > > > **(d).Do performance improvements actually happen on instances that are more "visual"?**
> > > > >
> > > > > We show the visually-hungry words (VH-words) extracted by syntax-, attention- and learning-based strategies in the Table 12, Table 13, Table 14, and Table 15 of our manuscript. The cases show that most of the extracted VH-words are generally related to some visual semantics, e.g., spider, two eyes. Although such VH-words can not perfectly cover all the visual semantics, they actually contain most of the important words that the PLMs may be hungry for, e.g., red and yellow. As the above response to **(c)**, our proposed method can filter out the redundant visual information and keep important visual information that PLM is hungry for. Hence VH-words do not only contain explicit visual information, such as yellow and small, but also implicit visual information, such as walk and waterfall.  And the extracted VH-words may not always align with human intuition.

---

> > > ### Author Response · Authors · 2022-12-06
> > > **The new response to the remaining concerns of the reviewer.**
> > >
> > > **We sincerely thank the reviewer for the updated comments and score. And we list our new response to the reviewer’s concerns with experimental results and analyses as follows.**
> > >
> > > **c.The effect of freezing the text encoder of VL-PTMs.**
> > > Based on the above two experiments, we can see that the text encoder of CLIP-base learned visual information to some extent. The results indicate that cross-modal contrastive objective and image-text corpus are both the main reasons for CLIP-base as a visual representation argumentation tool in our approach. Hence we freeze the text encoder of CLIP during fine-tuning on downstream tasks. To verify its effectiveness, we tried to update the text encoder of VL-PTMs during fine-tuning. We conduct experiments on commonsense reasoning and NLU tasks, and the results are shown in the following table. As we can see, even with more trainable parameters, the variation leads to a slight performance decrease. It further indicates that visual knowledge of CLIP is self-contained. Hence CLIP does not need to adapt to downstream tasks via updating parameters and can be used directly. Besides, we notice a similar performance decrease when replacing CLIP-base with ALBEF in our approach. It further indicates that the text encoder of VL-PTMs is a knowledge resource in our proposed method, i.e., it can be viewed as a reorganized memory containing visual knowledge from the large scale of the image-text corpus. With the visual knowledge memory, our proposed approach boosts the improvement of PLMs.
> > >
> > > \\begin{array} {|c|c|c|c|c|c|c|}
> > > \\hline
> > >  \\textrm{The text encoder of different VL-PTMs~(Params)} &\\textrm{Update the text encoder of VL-PTMs} & \\textrm{CSQA-3k} & \\textrm{CSQA}         & \\textrm{SST-2}  & \\textrm{STS-B}  & \\textrm{MNLI} \\\\
> > > \\hline
> > > \\textrm{ALBEF \(110M\)}      &×          &63.64	&68.47	&90.72	&87.17	&80.86 \\\\
> > > \\hline
> > >  \\textrm{ALBEF \(110M\)}    &√          & 62.46	&67.48	&89.79	&86.53	&80.17 \\\\
> > > \\hline
> > > \\textrm{CLIP-base \(52M\)}     &×          & \textbf{65.10}   &  \textbf{71.07}       &  \textbf{91.41}  & \textbf{87.73} & \textbf{82.27} \\\\
> > > \\hline
> > > \\textrm{CLIP-base \(52M\)}      &√        & 62.17	&69.20	&90.25	&86.60	&82.34 \\\\
> > > \\end{array}
> > >
> > > According to the quantitative experiments above, we can see that the text encoder of CLIP indeed provides visual knowledge that PLMs are hungry for. To further verify it, we also add a qualitative analysis to evaluate whether the augmented visual information aligns with human intuition. According to the case study in Table 12, Table 13, Table 14, and Table 15 of our manuscript, we can see that the extracted VH-words can cover most visual semantics of input text. It indicates that our proposed method supplies visual information for PLM via VH-words. And without the augmented visual knowledge, it seems difficult for PLMs to fully understand these VH-words.
> > >
> > > **In addition to the above experiments, we are still very willing to continue discussing with you for fully resolving your concerns. If you also have any other suggestions or questions, please feel free to let us know. We will continue to try our best to answer for you.**

---

> > > ### Author Response · Authors · 2022-12-12
> > > **Reminder for the Discussion**
> > >
> > > Dear reviewer TuCV,
> > >
> > > We want to send you a friendly reminder that the second stage of discussion will be completed soon. **We have added new experiments and analyses on your remaining concerns.** We are still very willing to discuss this with you in order to fully address your concerns. Please feel free to contact us with any additional suggestions or questions. We will keep doing our best to provide you with answers.

---

> ### Author Response · Authors · 2022-11-19
> **Response to the concerns about the VH-words extraction strategies and the analysis of performance gains on different tasks.**
>
> **Q4. Which words does the model select as "visually hungry"? Do they align with the syntax-based method? Do they align with human judgments of visual-ness?**
>
> In the response to Q1.b) of the reviewer 4pzW, we have shown the visually-hungry words (VH-words) extracted by the syntax-based strategy. The cases show that most of the extracted VH-words are generally related to some visual semantics, e.g., spider, two eyes. Here, we follow the suggestion of the reviewer and further show the extracted VH-words of the attention- and learning-based strategies as follows. We can see that the three strategies would extract slightly different VH-words. The reason is that the three strategies are based on different techniques to identify the VH-words. Whereas, these strategies all cover most of the important visual related words, e.g., two eyes and waterfall. Besides, we can also see that the VH-words extracted by our three strategies may not perfectly align with human judgment. In fact, it is also hard for humans to determine proper rules to identify VH-words, e.g., people, human, and water. Besides, as the learned knowledge of PLM is a black box, it is also difficult for humans to judge the usefulness of our extracted VH-words for PLMs. Therefore, the improvement of our approach on PLMs in Table 1,2,3,4 may be more persuasive to verify that our approach indeed augments useful visual knowledge to improve PLMs.
>
> **Input sentence:**
> Unlike a spider and his many sight seers, people only have what? two eyes. (CSQA)
>
> **Syntax-based strategy:**
> Unlike a spider and **his many sight seers**, **people** only have what? **two eyes**.
>
> **Attention-based strategy:**
> Unlike a **spider** and his many **sight seers**, **people** only have what? **two eyes**.
>
> **Learning-based strategy:**
> Unlike a **spider** and his many **sight seers**, people only have what? **two eyes**.
>
> **Input sentence:**
> Where on a river can a human hold a cup upright to catch water on a sunny, clear day? waterfall. (CSQA)
>
> **Syntax-based strategy:**
> Where on a **river** can a **human** hold a **cup** upright to catch **water** on a **sunny, clear day**? **waterfall**.
>
> **Attention-based strategy:**
> Where **on** a **river** can a **human** hold a **cup upright** to catch **water** on a sunny, clear day? **waterfall**.
>
> **Learning-based strategy:**
> Where on a **river** can a human **hold** a **cup upright** to **catch water** on a sunny, clear day? **waterfall**.
>
>
> **Q5. What kinds of tasks your model can outperform the baselines the most? Are these cases where we would expect visual information to help?**
>
> As shown in the response to Q4 of the reviewer A4xX, our proposed approach brings more significant improvement on smaller datasets (e.g., MRPC) and commonsense reasoning task (e.g., CSQA). The reason is that these datasets are relatively smaller and much more hungry for commonsense knowledge information, especially visual commonsense knowledge. Actually, such a result is as what we expect, since the key motivation of our approach is to augment the useful visual information to improve PLMs during fine-tuning. In the scenarios where text information is insufficient and visual commonsense is hungry for, our approach seems to be a promising solution to enhance PLM for achieving satisfactory performance.

---

### Official Review · Reviewer_A4xX · 2022-10-23

**Confidence:** 4
**Correctness:** 3
**Technical Novelty And Significance:** 3
**Empirical Novelty And Significance:** 2
**Recommendation:** 5

**Clarity, Quality, Novelty And Reproducibility:**

The paper is easy to follow as their proposed methods are relatively simple.

The idea of not using either retrieved or generated images for integrating visual information is novel and interesting, as it has been pointed out by the paper that sometimes the original text representations can be polluted using previous methods.

They claim that their code, data, and models will be publicly available.



**Strength And Weaknesses:**

Strengths:
1. The paper focuses on how to improve the model performance on pure-language tasks by introducing visual information, which is an important research direction yet has not been widely studied yet.
2. The idea of not using either retrieved or generated images for integrating visual information is novel and interesting, as it has been pointed out by the paper that sometimes the original text representations can be polluted using previous methods.
3. They conduct experiments across various tasks and demonstrate improvements over their baselines.

Weaknesses:
1. Their ways of extracting "visually-hungry" words are not technically sound and sometimes even arbitrary to me. For example, for their syntax-based strategy, they simply use an external tool to extract all the nouns and adjectives from sentences. As mentioned in the paper, not all nouns and adjectives are related to visual semantics, and using external tools can cause error propagation issues, yet this is their default strategy for some tasks as the other methods is computationally more costly or achieves similar performance. More technically sound approaches should be proposed, and the authors should conduct qualitative and/or quantitative analyses on the extracted VH words.
2. Their evaluation settings are questionable. It is unclear why they chose to report numbers on 6 GLUE tasks whereas most previous work would report results on 8 or 9 GLUE tasks. In addition, for GLUE tasks, they use a relatively weak baseline from Tan et al. [2020], but they should also include the baselines from the original BERT/RoBERTa papers and build their models on top of them so that the readers can be more clear about how their models perform.
3. There are few analyses on why their model can achieve improvements over pure-text models. They should include more analyses in this regard. For example, they can include ablation studies on whether the improvements are because of ensembling different text encoders, on what kinds of sentences/tasks their model can outperform the baselines the most, what if there are no visually-hungry words or their models extract wrong VH words, etc.

**Summary Of The Paper:**

The paper proposes to integrate visual information into pre-trained language models during fine-tuning when applied to pure-language tasks. Different from previous work that either uses retrieved or generated images, they propose to leverage the CLIP text encoder to obtain the image-aligned text representations of certain input words, and then feed the language models with these additional representations when fine-tuned on language tasks. They demonstrate improvements over baselines on text classification, commonsense classification/generation, and visual entailment tasks.

Specifically, they propose three strategies to extract "visually-hungry words" (VH words): 1) using spacy to extract the POS of each word and treating nouns and adjectives as VH words; 2) using the attention scores of the CLIP text encoder and selecting the top-k words with the highest attention scores with the EOS token; 3) training a neural network with Gumbel-Softmax to select the VH words automatically.

After they extract the VH words, they use the CLIP text encoder with prompts to get the image-aligned representations of these words and feed them to their language models.

**Summary Of The Review:**

Overall, while the idea of using an image-aligned text encoder to provide visual information for language tasks is interesting and may have potential, the currently proposed methods are not technically sound and the evaluation settings are questionable (the details of these are listed in "strength and weakness"), therefore, I am leaning towards a rejection of the paper.

---

> ### Author Response · Authors · 2022-11-19
> **Response to the concerns about the analysis of VH-words and supplementary results on GLUE.**
>
> We sincerely thank the reviewer for the insightful suggestions. We have carefully revised our manuscript and listed the detailed responses to the concerns in the reviewer's suggestions.
>
> **Q1.The quantitative and qualitative analysis of visually-hungry words.**
>
> We show the visually-hungry words (VH-words) extracted by the syntax-based strategy as follows. The cases show that most of the extracted VH-words are generally related to some visual semantics, e.g., spider, two eyes. Although such VH-words can not perfectly cover all the visual semantics, they actually contain most of the important words that the PLMs may be hungry for, e.g., red and yellow. In addition to the case study, we also conduct analysis experiments to show how the quality of the VH-words affects the performance of our approach. As shown in the following table, we can see that improper VH-words will significantly degrade the model performance.
>
> 1.Unlike a **spider** and his **many sight seers**, **people** only have what? two eyes. (CSQA)
>
> 2.Where do you find **the most amount of leafs**? **forrest**. (CSQA)
>
> 3.that **warm water** under a **red bridge** is a **poem** to the enduring strengths of **women**. (SST-2)
>
> 4.the **mesmerizing performances** of the leads keep the **film grounded** and keep the **audience riveted**. (SST-2)
>
> 5.How do I sell dry Moringa **leaves powder** in Indian **market**?   Can I use the moringa **leaves** that are already starting to turn **yellow** or **yellowish**? (QQP)
>
>
> \\begin{array} {|c|c|c|c|}
> \\hline
>  \\textrm{The proportion of correct visual words} & \\textrm{CSQA-3k}       & \\textrm{SST-2}   & \\textrm{QQP} \\\\
> \\hline
> \\textrm{0 \\% }  & 61.60                   &  89.57           & 87.63 \\\\
> \\hline
> \\textrm{20 \\% }& 62.17                   &  89.44           & 87.40   \\\\
> \\hline
> \\textrm{50 \\% }                & 64.22                   &  91.73           & 89.20 \\\\
> \\hline
> \\textrm{100 \\%}              & \textbf{65.10}          &  \textbf{92.93}  & \textbf{89.74} \\\\
>  \\hline
> \\textrm{None}              & 61.88                   &  89.23           & 86.21 \\\\
> \\end{array}
>
> **Q2.a. It is unclear why they chose to report numbers on 6 GLUE tasks whereas most previous work would report results on 8 or 9 GLUE tasks.**
>
> In our approach, we follow the experimental setting of Voken and iACE for a fair comparison, hence we only report the results on 6 NLU tasks. Here, following the suggestion of the author, we also evaluate our proposed method on the other three tasks. The results are as follows. We can see that our proposed method can also consistently boost the performance of the PLMs.
>
> \\begin{array} {|c|c|c|c|}
> \\hline
>  \\textrm{Method}
>                       & \\textrm{WNLI}  & \\textrm{RTE}  &  \\textrm{CoLA} \\\\
> \\hline
> \\textrm{BERT}_\\textrm{base} & 60.6            & 75.0          & 50.5  \\\\
> \\hline
> \\textrm{BERT}_\\textrm{base}  \+ \\textrm{SBS }& 62.0            & 78.2            & 56.8 \\\\
> \\hline
> \\textrm{BERT}_\\textrm{base}  \+  \\textrm{VABS }         & \textbf{63.4}   & 79.0           & 57.1 \\\\
> \\hline
> \\textrm{BERT}_\\textrm{base}   \+  \\textrm{ LBS }       & \textbf{63.4}   & \textbf{79.7} & \textbf{57.4}
> \\end{array}
>
> \\begin{array} {|c|c|c|c|}
> \\hline
>  \\textrm{Method}
>                       & \\textrm{WNLI}  & \\textrm{RTE}  &  \\textrm{CoLA} \\\\
> \\hline
> \\textrm{RoBERTa}_\\textrm{base} & 62.0          & 78.6            & 59.1  \\\\
> \\hline
> \\textrm{RoBERTa}_\\textrm{base}  \+  \\textrm{SBS} & 63.4	      & 80.4	        & 60.3	 \\\\
> \\hline
> \\textrm{RoBERTa}_\\textrm{base}  \+   \\textrm{VABS }       & \textbf{64.8} & \textbf{81.8}	        & 60.8 \\\\
> \\hline
> \\textrm{RoBERTa}_\\textrm{base}   \+   \\textrm{ LBS}     & \textbf{64.8} & \textbf{81.8}   & \textbf{61.0}
> \\end{array}

---

> ### Author Response · Authors · 2022-11-19
> **Response to the concerns about the supplementary results on GLUE, ablation study, and the effect of the bad VH-words.**
>
> **Q2.b. They should also include the baselines from the original BERT/RoBERTa papers and build their models on top of them.**
>
> For the results in Table 1, we adopt the same experimental settings as iACE and Voken for a fair comparison, including the settings of the pre-trained models (i.e., BERT and RoBERTa). Following the suggestion of the reviewer, we utilize the same experimental settings as the original BERT/RoBERTa papers and conduct new experiments in NLU tasks. The results are as follows. It is obvious that our VAWI also performs consistently better than the original RoBERTa-base on the 6 datasets. It further indicates the effectiveness of our approach.
>
> \\begin{array} {|c|c|c|c|}
> \\hline
>  \\textrm{Method}
>                       &  \\textrm{SST-2} & \\textrm{QNLI}  &  \\textrm{QQP} & \\textrm{MNLI} & \textbf{MRPC} & \textbf{STS-B} \\\\
> \\hline
> \\textrm{RoBERTa}_\\textrm{base}  & 94.8           & 92.2           & 91.9          & 87.9          & 90.2          & 91.2   \\\\
> \\hline
> \\textrm{RoBERTa}_\\textrm{base}  \+ \\textrm{SBS} & 95.4	         & 92.9	          & 92.6	      & 89.0	      & 91.0	      & 91.9\\\\
> \\hline
> \\textrm{RoBERTa}_\\textrm{base}  \+  \\textrm{VABS}        & \textbf{95.8}  & 93.0	          & \textbf{92.8} & \textbf{89.3} & 90.9	      & \textbf{92.3}\\\\
> \\hline
> \\textrm{RoBERTa}_\\textrm{base}   \+  \\textrm{ LBS}        & 95.6           & \textbf{93.1}   & \textbf{92.8} & 89.1	      & \textbf{91.2} & \textbf{92.3}
> \\end{array}
>
> **Q3. Ablation studies on whether the improvements are because of ensembling different text encoders.**
>
> We have conducted the ablation and variation study of our approach by using different text encoders to replace our CLIP-base. As shown in Table 5 of our manuscript, we can see that all these variations (even using Billion-scale T5) perform not better than our approach. It indicates that CLIP-base is more effective to augment visual knowledge to improve the performance of PLMs, as it has pre-learned the visual knowledge from large-scale image-text pairs. Besides, in our response to Reviewer YcbG Q1, we also further investigate the effectiveness of CLIP-base by adding more ablation experiments, using stronger VL-PTMs and using the PLM pre-trained on only captions.
>
> **Q4. What kinds of sentences/tasks your model can outperform the baselines the most?**
>
>  Following the suggestion of the reviewer, we comprehensively analyze the experimental results of our model in Table 1 and Table 2, and summarize the performance improvements of our approach on RoBERTa-base and T5-3B as follows. The results show that our proposed approach is more suitable for smaller datasets (e.g., MRPC) and commonsense reasoning task (e.g., CSQA), and brings more significant improvement on them. The reason is that these tasks are much more hungry for the complementary information, especially the visual knowledge information.
>
> \\begin{array} {|c|c|c|c|}
> \\hline
>  \\textrm{Base Model}
>                       &\textbf{SST-2} & \textbf{QNLI}  & \textbf{QQP} & \textbf{MNLI} & \textbf{MRPC} & \textbf{STS-B} &\textbf{CSQA-3K} &\textbf{CSQA}\\\\
> \\hline
> \\textrm{RoBERTa}_\\textrm{base}   & 2.2           & 1.9           & 1.5           & 3.2           & 6.8           & 2.3           &3.2               &3.1  \\\\
> \\hline
> \\textrm{T5-3b}      & 1.3           & 1.3           & 0.9           & 2.2           & 4.9           & 2.3           &2.1               &2.3
> \\end{array}
>
> **Q5.The effect of the bad visually-hungry words.**
>
> In the response to Q1, we have shown that bad visually-hungry words may affect the performance of our approach. Here, we further conduct experiments to show the effect of insufficient VH-words on our model performance. After extracting the VH-words, we remove part of them and only randomly sample 0%, 20%, and 50% VH-words for augmentation. As shown in the following table, we can see that with the decreasing of the sampling probability, the performance of our approach degrades gradually. It indicates that not enough VH-words would degrade the performance of our approach. As shown in the results, we can see that with the decreasing of the sampling probability, the performance of our approach degrades gradually.
>
> \\begin{array} {|c|c|c|c|}
> \\hline
>  \\textrm{The proportion of correct visual words} & \\textrm{CSQA-3k}       & \\textrm{SST-2}   & \\textrm{QQP} \\\\
> \\hline
> \\textrm{0 \\% }  & 61.60                   &  89.57           & 87.63 \\\\
> \\hline
> \\textrm{20 \\% }& 62.17                   &  89.44           & 87.40   \\\\
> \\hline
> \\textrm{50 \\% }                & 64.22                   &  91.73           & 89.20 \\\\
> \\hline
> \\textrm{100 \\%}              & \textbf{65.10}          &  \textbf{92.93}  & \textbf{89.74} \\\\
>  \\hline
> \\textrm{None}              & 61.88                   &  89.23           & 86.21 \\\\
> \\end{array}

---

> ### Author Response · Authors · 2022-12-07
> **Reminder for the Discussion**
>
> Dear Reviewer A4xX,
>
> We want to send you a friendly reminder that the second stage of discussion will be completed soon. Here are the things that we have added and resolved by your valuable feedback!
>
> - We fixed typos and unclear descriptions, and carefully revise the writing of our manuscript.
> - We add more ablation studies on the effect of the source of visual representations, the pre-trained dataset, and the stronger VL-PTMs.
> - We add the analysis experiments about computation latency introduced by our method.
> - We add the experiments about the effect of the improper visually-hungry words.
> - We add further analysis on the interpretability of augmented embeddings.
>
> Thanks for your willingness to reconsider your score based on our responses, and we really want to know whether our responses address your concerns. If there is any other concern that we could not address in the response, please feel free to let us know and we would be happy to provide further explanation.
>
> Thanks.

---

> > ### Comment · Reviewer_A4xX · 2022-12-08
> > **Response**
> >
> > Dear Authors,
> >
> > I sincerely appreciate the additional experiments and they are quite helpful. However, my major concern is still not addressed yet, which is that the proposed ways of extracting "visually-hungry" words are not technically sound and sometimes even arbitrary to me. As in the rebuttal, you have demonstrated that improper VH-words will significantly degrade the model performance. Therefore, this issue is even more concerning and as in my previous review, I would strongly encourage the authors to propose more technically sound approaches. It would also be helpful to manually collect such a dataset for identifying "visually-hungry" words and show the accuracy of your models.
> >
> > Best

---

> > > ### Author Response · Authors · 2022-12-09
> > > **The new response to the remaining concerns of the reviewer.**
> > >
> > > **We sincerely thank the reviewer for the updated comment. And we list our new response to the reviewer’s remaining concerns with two-fold analyses as follows.**
> > >
> > > - Visually hungry words (VH-words) are difficult to define and vary across different pre-trained language models (PLMs) and downstream tasks. Hence it is hard to annotate VH-words manually. Considering these issues, we propose three simple methods to extract VH-words and our experimental results show that these methods can achieve remarkable performance on 10 NLP tasks.
> > >
> > > - Our three proposed methods for extracting VH-words are based on different perspectives. The first two methods, the syntax-based strategy and the visually-enhanced attention-based strategy, are rule-based and more efficient. It is worth mentioning that the last method, the learning-based strategy, leverages the gradient-based optimization algorithms to update the parameters of the VH-words extraction module. Hence this method can adaptively extract VH-words to fit the need for different PLMs and specific tasks. And it seems that this method is more feasible and effective than the first two methods. However, as the results are shown in Table 1 of our manuscript, the three methods we evaluate on NLU tasks achieve similar performance gains. Hence, from our perspective, efficiency is more important for extracting VH-words in these three methods. As shown in Table 9 of our manuscript, the computational cost of the syntax-base strategy is remarkably lower than the other two strategies. For efficiency, we choose the syntax-based strategy as our default method for extracting VH-words. Therefore, based on these three methods, we have not done further exploration for extracting VH-words in this work. However, we will take your valuable suggestion into consideration to investigate more effective and efficient approaches for extracting VH-words in our future work.
> > >
> > > **We are still very willing to continue discussing with you for fully resolving your concerns. If you also have any other suggestions or questions, please feel free to let us know. We will continue to try our best to answer for you.**

---

> > > ### Author Response · Authors · 2022-12-12
> > > **Reminder for the Discussion**
> > >
> > > Dear reviewer A4xX,
> > >
> > > We want to send you a friendly reminder that the second stage of discussion will be completed soon. **We have added the analyses on your remaining concerns.**
> > > We are still very willing to discuss this with you in order to fully address your concerns. Please feel free to contact us with any additional suggestions or questions. We will keep doing our best to provide you with answers.

---

### Official Review · Reviewer_4pzW · 2022-10-25

**Confidence:** 4
**Correctness:** 3
**Technical Novelty And Significance:** 3
**Empirical Novelty And Significance:** 3
**Recommendation:** 6

**Clarity, Quality, Novelty And Reproducibility:**

The writing is clear and easy to understand.
Novelty is good and it should be easy to reproduce.

**Strength And Weaknesses:**

Strengths:
1. Compared to recent works of Visually-augmented PLM, this work doesn't need to retrieve or generate images, which is much faster in training/inference and more straightforward in methodology.
2. Experiment section covers many different datasets and previous methods. The proposed method can outperform others and baselines by a satisfactory margin.

Weaknesses:
1. Some detail of the visually-hungry word extraction is not clear to me.
- a) In the learning-based strategy, when back-propagation, will the VL-PTM (CLIP in your case) and PLM all be updated? Or just the MLP is updated?
- b) In those three strategies, will the visual bi-gram or phrase be detected one unit? For example, in the case of 'He is eating a green apple', 'green apple' should be detected as one phrase instead of two words. In the following CLIP text encoding, 'green apple' should also be encoded as in one sentence instead of encoding 'green' and 'apple' separately.
2. In experiments, why T5-3B model is missing in NLU tasks while it's used in other tasks?
3. The details about the experimental setting of `+retrieved images` in Tab2, 3 are missing.  What's the image database? How many images are retrieved for each VH token? How is it different from other retrieving-based methods such as iACE?

**Summary Of The Paper:**

This paper proposed to inject visual information into pre-trained language models without retrieved or generated images. Instead, it detects visually-hungry words and generates their visual representation by CLIP text encoder, and injects them back into a pre-trained language model. Experiments are done in various datasets and recent methods are compared.

**Summary Of The Review:**

Overall, I like the straightforward idea and it also works well. Some details and experiments are missing but I still lean to accept it.

---

> ### Author Response · Authors · 2022-11-19
> **Response to the concerns about the more details in our method and supplementary results of T5-3b.**
>
>
> We sincerely thank the reviewer for the insightful suggestions. We have carefully revised our manuscript and listed the detailed responses to the concerns in the reviewer's suggestions.
>
> **Q1. More detail in visually-hungry words extraction.**
>
> Thanks for your careful reading. We have refined the writing of this part to clearly describe our approach.
>
> **a.In the learning-based strategy, which part of the model is updated?**
>
> We only update the MLP layer in the learning-based strategy, and freeze the PLM and the VL-PTM for efficiency.
>
> **b.In those three strategies, will the visual bi-gram or phrase be detected as one unit?**
>
> We detect the visual phrase in the three strategies. For the syntax-based strategy, as the phrases can be generally annotated by the syntactic analysis toolkit, we directly pick them as a unit. For the attention- and learning-based strategies, we collect the neighbouring VH-words to compose the phrases. Note that all the above ways may also select improper visual words or phrases by mistake, hence we also incorporate the reformulation layer with cross-attention mechanism to adaptively capture and fuse their visual knowledge.
>
> **Q2. The results of T5-3b in NLU tasks.**
>
> For NLU tasks, we follow the experimental settings of Voken and iACE for a fair comparison. As Voken and iACE only train RoBERTa-base and BERT-base, we do not consider using T5-3b. Here, following the suggestion of the reviewer, we also implement our proposed method on T5-3b for NLU tasks. The results are as follows. We can see that our approach can also achieve remarkable performance on 6 NLU tasks based on T5-3b. It further shows the effectiveness of our approach in NLU tasks and large-scale PLMs.
>
> \\begin{array} {|c|c|c|c|}
> \\hline
>  \\textrm{The text encoder of different VL-PTMs~(Params)} & \\textrm{SST-2} & \\textrm{QNLI}  & \\textrm{QQP} & \\textrm{MNLI} & \\textrm{MRPC} & \\textrm{STS-B} &\\textrm{Avg.} \\\\
> \\hline
> \\textrm{+None}  & 91.4           & 90.2           & 89.7          & 84.3          & 85.6          & 88.4          & 88.26 \\\\
> \\hline
> \\textrm{+VAWI-SBS}& 92.7	         & 91.5	          & 90.6	      & 86.5	      & 90.5	      & 90.7	      & 90.41\\\\
> \\hline
> \\textrm{+VAWI-VABS}                 & 92.7           & 91.6	          & \textbf{90.9} & \textbf{86.9} & 90.5	      & \textbf{90.9} & 90.58 \\\\
> \\hline
> \\textrm{+VAWI-LBS}              & \textbf{92.9}  & \textbf{91.9}  & 90.8          & 86.8	      & \textbf{90.7} & \textbf{90.9} & \textbf{90.66}
> \\end{array}
>
> **Q3. The details about the retrieved images in commsenseQA and CommonGen.**
>
> The experiments about +retrieved images are to show if the CLIP text encoder is more effective than an intuitive way that first retrieves the images about the VH-words and then encodes them by image encoder. Therefore, we first extract VH-words using the same strategy as our approach, and then use them to retrieve eight most relevant images. Considering that the existing image-text dataset may not always contain the relevant images, we retrieve the images using a widely-used search engine, Google Images. After searching the top-ranked images, we also utilize CLIP to filter the irrelevant images for guaranteeing image-text relevance. We have also checked the selected images in this way and found the images are mostly relevant to the visually-hungry words. After that, we also utilize the CLIP image-encoder to encode these images and insert them into the PLM as the prefix. As shown in Table 2,3, we notice that our proposed method outperforms this variation using these high-quality images. It indicates that our ''without image'' approach is more effective than such a complicated way using retrieved images.

---

> ### Author Response · Authors · 2022-12-07
> **Reminder for the Discussion**
>
> Dear Reviewer 4pzW,
>
> We want to send you a friendly reminder that the second stage of discussion will be completed soon. Here are the things that we have added and resolved by your valuable feedback!
>
> - We fixed typos and unclear descriptions, and carefully revise the writing of our manuscript.
> - We add more ablation studies on the effect of the source of visual representations, the pre-trained dataset, and the stronger VL-PTMs.
> - We add the analysis experiments about computation latency introduced by our method.
> - We add the experiments about the effect of the improper visually-hungry words.
> - We add further analysis on the interpretability of augmented embeddings.
>
> Thanks for your willingness to reconsider your score based on our responses, and we really want to know whether our responses address your concerns. If there is any other concern that we could not address in the response, please feel free to let us know and we would be happy to provide further explanation.
>
> Thanks.

---

> ### Author Response · Authors · 2022-12-12
> **Reminder for the Discussion**
>
> Dear reviewer 4pzW,
>
> We want to send you a friendly reminder that the second stage of discussion will be completed soon. Here are the things that we have added and resolved by your valuable feedback!
>
> **1. We add more details in visually-hungry words extraction and the retrieved images.**
> **2. The case study of visually-hungry words extracted by our three proposed methods.**
> **3. New experiments of our proposed method on T5-3b for NLU tasks.**
>
> Thanks for your willingness to reconsider your score based on our responses, and we really want to know whether our responses address your concerns. If there is any other concern that we could not address in the response, please feel free to let us know and we would be happy to provide further explanation.

---

### Official Review · Reviewer_YcbG · 2022-10-31

**Confidence:** 4
**Correctness:** 3
**Technical Novelty And Significance:** 2
**Empirical Novelty And Significance:** Not applicable
**Recommendation:** 5

**Clarity, Quality, Novelty And Reproducibility:**

Clarity, quality and reproducibility look overall good. The technical part is not entirely novel but certainly interesting.

**Strength And Weaknesses:**

Pros:
1. The proposed idea is quite intuitive and straightforward.
2. The evaluation is extensive on ten NLP tasks.

Cons:
1. Insufficient analysis to identify the true contribution of performance gain.
a. The baselines in Table 5 should be actually included for the other evaluations as it is important to know whether the proposed method is only significantly effective on other tasks/datasets.
b. It is unclear whether stronger VL models lead to better performance when used in the proposed method.
c. Most importantly, despite the performance gain of CLIP text encoder as reported in Table 5, it is unclear whether this really comes from "seeing the visual world". What if the knowledge source actually comes from the captions? Therefore, it is important to add baselines to justify this problem. For example, using image captions to pretrain a language model as the knowledge source; retrieve some image captions as the knowledge source.
2. Additionally, could the augmented embeddings be interpreted? Are there any evidence from the interpretation really also supporting the claims?
3. The extra computation and latency introduced by the proposed method should be also provided to help the audience understand better the trade-off between visual augmentation and efficiency.


Minor:
Second to last paragraph in introduction,  "improveing".

**Summary Of The Paper:**

In this paper, the authors proposed to leverage pretrained vision-language models like CLIP to augment the pretrained language models with visual knowledge by injecting embeddings generated from the text encoder in CLIP for words that may be lack of visual knowledge. Effectiveness of the propopsed method is verified on various benchmarks on natural language understanding, commonsense reasoning, text generation and visual entailment.

**Summary Of The Review:**

Although the reviewer thinks the proposed idea might be interesting to the community, the analysis on the real source of performance gain is not enough to really support the claim.

---

> ### Author Response · Authors · 2022-11-19
> **Response to the concerns about the true contribution of performance gain.**
>
> We sincerely thank the reviewer for the insightful suggestions. We have carefully revised our manuscript and listed the detailed responses to the concerns in the reviewer's suggestions.
>
> **Q1. More analysis to identify the true contribution of performance gain.**
>
> **a. The baselines in Table 5 should be actually included for the other evaluations.**
>
> Table 5 shows the results of our ablation and variation study. We choose three representative tasks including commonsense reasoning and two NLU tasks, whose dataset scales vary from 2.9k to 363.8k. Here, following the suggestion of the reviewer, we also add new ablation and variation experiments on other four tasks, including commonsense reasoning (CSQA), text generation (CommonGen), and two NLU tasks (STS-B and QNLI). The results are shown as follows. We can see that our approach is consistently better than all these variants in the four tasks. It further indicates that CLIP-base is more effective to augment visual knowledge to improve the performance of PLMs.
>
> \\begin{array} {|c|c|c|c|}
> \\hline
>  \\textrm{Source of visual representation~(Params)}& \\textrm{CSQA} & \\textrm{STS-B} &\\textrm{QNLI} \\\\
> \\hline
> \\textrm{Random Noise \(0M\)}  & 67.78 &85.13 &87.22  \\\\
> \\hline
> \\textrm{RoBERTa-Large \(355M\)} & 67.17 &85.60 &87.77 \\\\
> \\hline
> \\textrm{T5-large-encoder \(375M\)} & 67.87 &86.40 &87.94 \\\\
> \\hline
> \\textrm{T5-3b-encoder \(1500M\)} & 68.42 &86.93 &88.21 \\\\
> \\hline
> \\textrm{CLIP-base \(52M\)} &  \textbf{71.07} & \textbf{87.73} & \textbf{89.40} \\\\
> \\end{array}
>
> \\begin{array} {|c|c|c|c|}
> \\hline
>  \\textrm{Source of visual representation~(Params)}& \\textrm{BLUE-3} & \\textrm{BLUE-4} &\\textrm{METOR} &\\textrm{Rouge-L}&\\textrm{CIDER}&\\textrm{METOR} \\\\
> \\hline
> \\textrm{Random Noise \(0M\)}  & 42.76           & 32.27             & 31.41          & 57.32            & 16.37          & 32.77  \\\\
> \\hline
> \\textrm{RoBERTa-Large \(355M\)} & 42.82           & 32.87             & 31.65          & 57.39            & 16.42          & 33.12 \\\\
> \\hline
> \\textrm{T5-large-encoder \(375M\)} & 42.89           & 32.86             & 31.74          & 57.42            & 16.59          & 33.08 \\\\
> \\hline
> \\textrm{T5-3b-encoder \(1500M\)} & 43.21           & 33.14             & 31.85          & 57.52            & 16.72          & 33.19 \\\\
> \\hline
> \\textrm{CLIP-base \(52M\)} & \textbf{44.56}  & \textbf{34.17}    & \textbf{32.47} & \textbf{58.46}   & \textbf{17.23} & \textbf{33.67}\\\\
> \\end{array}
>
> **b. It is unclear whether stronger VL models lead to better performance.**
>
> In our work, we choose CLIP-base to enhance PLMs, as it has been pre-trained on a large-scale image-text dataset. Generally, a stronger VL-PTM would be more promising to further improve the performance. Here, we follow the suggestion of the reviewer and replace our CLIP-base model with some stronger VL-PTMs, e.g., ALBEF, UniCL-base, and CLIP-large. Concretely, ALBEF leverages more pre-training tasks (e.g., MLM, ITM, and ITC), UniCL utilizes more high-quality pre-training data, and CLIP-large increases the scale of model parameters. We also evaluate the above variations on CSQA-3k, QQP, and SST-2, and the results are as follows. We can see that UniCL and CLIP-large outperform CLIP-base. It indicates that the VL-PTM with a larger scale of model parameters or more high-quality pre-training data is more capable of augmenting useful visual knowledge for PLMs. Whereas considering the efficiency, CLIP-base is also a good choice in our proposed approach, and we will investigate more proper VL-PTMs in the future.
>
> \\begin{array} {|c|c|c|c|}
> \\hline
>  \\textrm{The text encoder of different VL-PTMs~(Params)}& \\textrm{CSQA-3k} & \\textrm{SST-2} &\\textrm{QQP} \\\\
> \\hline
> \\textrm{Random Noise \(0M\)}  & 61.59              &  89.23           & 86.21  \\\\
> \\hline
> \\textrm{ALBEF \(110M\)}& 63.34              &  90.72           & 87.17  \\\\
> \\hline
> \\textrm{CLIP-base \(52M\)} & 65.10              &  91.41           & 87.72 \\\\
> \\hline
> \\textrm{UniCL-base \(52M\)} & 65.98              &  91.75           & 88.07 \\\\
> \\hline
> \\textrm{CLIP-large \(123M\)}  & \textbf{66.27}     &\textbf{92.10}    &\textbf{88.31}  \\\\
> \\end{array}

---

> ### Author Response · Authors · 2022-11-19
> **Response to the concerns about the true contribution of performance gain, the interpretability of augmented embeddings and the extra latency.**
>
> **Q1. More analysis to identify the true contribution of performance gain.**
>
> **c. Whether knowledge is derived from captions.**
>
> Following the suggestion of the reviewer, we add new experiments that pre-train a new PLM only using the captions of the images. Following the setting of ALBEF, we utilize the pre-trained parameters of BERT to initialize this model and only extract the captions from the pre-training corpus of ALBEF (totally 14.5M sentences). After pre-training until convergence, we utilize this model to replace CLIP-base in our approach and keep other settings unchanged. We conduct experiments on commonsense reasoning and NLU tasks to evaluate its effectiveness for augmenting visual knowledge. The results are as follows. We can see that such a variation underperforms ALBEF and our approach, and even leads to performance degradation on the CSQA task. It indicates that the image data during pre-training is an important resource for learning visual knowledge in VL-PTMs. Only text data (i.e., captions) can not provide sufficient visual knowledge that PLMs are hungry for. Therefore, after pre-learned on large-scale text-image pairs, CLIP can absorb the useful visual knowledge from the images and inject them into PLMs in our approach.
>
> \\begin{array} {|c|c|c|c|}
> \\hline
>  \\textrm{The text encoder of different VL-PTMs~(Params)} & \\textrm{CSQA-3k} & \\textrm{CSQA}         & \\textrm{SST-2}  & \\textrm{STS-B}  & \\textrm{MNLI} \\\\
> \\hline
> \\textrm{Noise \(0M\)}  & 61.59            & 67.90                &  89.23           & 85.46          & 79.06 \\\\
> \\hline
> \\textrm{BERT pre-trained on captions \(110M\)}& 62.17           &  67.56                &  89.58       & 85.73          & 79.24   \\\\
> \\hline
> \\textrm{ALBEF \(110M\)}                 & 63.64            & 68.47               &  90.72           & 87.17          & 80.86 \\\\
> \\hline
> \\textrm{CLIP-base \(52M\)}              & \textbf{65.10}   &  \textbf{71.07}       &  \textbf{91.41}  & \textbf{87.73} & \textbf{82.27}
> \\end{array}
>
> **Q2. The interpretability of augmented embeddings.**
>
> Following the suggestion of the reviewer, we have added the attention visualization of our augmented embeddings in our new manuscript, to show how our augmented embeddings infuse the visual knowledge into the PLM. Concretely, we show the attention distributions of a PLM (i.e., RoBERTa-base) in the last few layers before and after infusing visually-augmented representations on CSQA. As shown in Table 11 of our new manuscript, we can see that the [CLS] tokens pay more attention to the visually-hungry words (VH-words) and their visually-augmented representations, and the VH-words also pay more attention to their visually-augmented representations. It shows that the injected visually-augmented representations indeed provide useful knowledge, which guides the PLM to focus on more important tokens and also improves the representations of the VH-words and the [CLS] token.
>
> **Q3. The extra computation and latency introduced by the proposed method.**
>
> In our proposed approach, we fix the model parameters of CLIP-base to preserve the visual knowledge. Such a way can also decrease the computation costs during training and inference. To verify it, following the reviewer's suggestion, we report the mean training and inference latency per batch on the CSQA-3k dataset of our method and baselines on RTX3090 GPU, where all these methods utilize RoBERTa-base as the backbone. As shown in the following table, we can see that our proposed VAWI-SBS and VAWI-ABS would not increase the latency too much. For VAWI-LBS, as it requires a PLM and a VL-PTM to adaptively select the VH-words, it will relatively increase the latency much. As shown in Table 1 of our manuscript, we can see that all the three variations achieve comparable performance in 6 NLU datasets. Therefore, it is more efficient and effective to select the SBS and ABS variations in our approach. Despite it, we can see that all our variations own less latency than iACE since our approach does not require a time-consuming image generation process. And as shown in Table 1 of our manuscript, our approach can also achieve better performance.
>
> \\begin{array} {|c|c|c|c|}
> \\hline
> \\textrm{Computation Costs} & \\textrm{RoBERTa-base}     & \\textrm{VAWI-SBS}       & \\textrm{VAWI-ABS }    & \\textrm{VAWI-LBS}     &\\textrm{Voken}     &\\textrm{iACE} \\\\
> \\hline
> \\textrm{Training Time \(s\)}  &0.506        &  0.587    & 0.680   & 0.893   & 0.506    & 1.138\\\\
> \\hline
> \\textrm{Inference Time \(s\)}&0.182        &  0.241    & 0.308   & 0.486   & 0.182    & 0.512   \\\\
> \\end{array}
>
>
> **Q4: Writing.**
>
> Thanks for your careful reading. We have fixed the mentioned problems and carefully revised our manuscript.

---

> ### Author Response · Authors · 2022-12-07
> **Reminder for the Discussion**
>
> Dear Reviewer YcbG,
>
> We want to send you a friendly reminder that the second stage of discussion will be completed soon. Here are the things that we have added and resolved by your valuable feedback!
>
> - We fixed typos and unclear descriptions, and carefully revise the writing of our manuscript.
> - We add more ablation studies on the effect of the source of visual representations, the pre-trained dataset, and the stronger VL-PTMs.
> - We add the analysis experiments about computation latency introduced by our method.
> - We add the experiments about the effect of the improper visually-hungry words.
> - We add further analysis on the interpretability of augmented embeddings.
>
> Thanks for your willingness to reconsider your score based on our responses, and we really want to know whether our responses address your concerns. If there is any other concern that we could not address in the response, please feel free to let us know and we would be happy to provide further explanation.
>
> Thanks.

---

> ### Author Response · Authors · 2022-12-12
> **Reminder for the Discussion**
>
> Dear reviewer YcbG,
>
> We want to send you a friendly reminder that the second stage of discussion will be completed soon. Here are the things that we have added and resolved by your valuable feedback!
>
> **1. We fix the mentioned typos, carefully read and revise the writing of our paper.**
> **2. New ablation studies to evaluate the effect of the source of visual representations, the pre-trained dataset, the contrastive objective, and the stronger VL-PTMs.**
> **3. New experiment to analyze the computation costs introduced by our method.**
> **4. New analyses on the interpretability of augmented embeddings.**
>
> Thanks for your willingness to reconsider your score based on our responses, and we really want to know whether our responses address your concerns. If there is any other concern that we could not address in the response, please feel free to let us know and we would be happy to provide further explanation.

---

### Author Response · Authors · 2022-11-19
**New Revision of our Manuscript**

We sincerely thank the four reviewers for their insightful and constructive feedback. We have provided a separate response to each reviewer and also updated the paper following the revision suggestions of the reviewers. We list the main revision content as follows:

We fixed the reviewer's mentioned typos and unclear descriptions, and carefully revise the writing of our manuscript.

We add more ablation studies on visual knowledge augmentation in Appendix B.1. Concretely, we further evaluate the effect of pretrained dataset and the stronger VL-PTMs, and show the experimental results on Table 6 and Table 7. It further indicates that the improvement of our method derives from the augmentation of the visual Knowledge about the VH-words.

We add the analysis experiments about computation latency introduced by our method in Appendix C, and show the results on Table 9. We can see that it is more efficient and effective to select the SBS and ABS variations in our approach.

We add the experiments about the effect of the improper visually-hungry words in Appendix C, and show the results in Table 10. We can see that with the decreasing of the sampling probability, the performance of our approach degrades gradually.

We further analyze the interpretability of Augmented Embeddings in Appendix C, and show the attention distributions of a PLM (i.e., RoBERTa-base) in the last layers before and after infusing visually-augmented representations. We can see that the [CLS] tokens pay more attention to the visually-hungry words (VH-words) and their visually-augmented representations, showing that the infused visual knowledge guides the PLM to focus on more important tokens.

We add case studies about our three VH-words extraction strategies in Appendix C, and the results are in Table 12, Table 13, Table 14 and Table 15. We can see that most of the extracted VH-words by our strategies are generally related to some visual semantics.

---

### Decision · Program_Chairs · 2023-01-20

**Decision:**

Reject

**Justification For Why Not Higher Score:**

The reviewers had a very informative discussion of this work and reached a consensus that there are still many experiments to add in order to support the claims of this paper, despite that the authors had made a significant revision during the discussion.

**Justification For Why Not Lower Score:**

N/A

**Metareview: Summary, Strengths And Weaknesses:**

This paper proposes a new method to augment pure language models with visual information to improve their performance on pure language tasks. Different from previous methods relying on image retrieval or generation, the proposed method leverages the CLIP text encoder to obtain the image-aligned text representations of "visually-hungry" words, and then feeds the language models with these representations when fine-tuning them on pure language tasks. The method is well-motivated and more efficient than previous methods, and the authors evaluate it on various tasks and datasets. However, reviewers raised common concerns regarding the definition of "visually-hungry" words, unfair evaluation settings, and lack of essential baselines. For example, the current experimental results cannot guarantee that the improvements actually come from visual augmentation. Most of those concerns still remain unresolved after the rebuttal and discussion.